# Zebrafish xenografts as a fast screening platform for bevacizumab cancer therapy

Cátia Rebelo de Almeida[1], Raquel Valente Mendes[1], Anna Pezzarossa[1], Joaquim Gago [2], Carlos Carvalho[2], António Alves [3], Vitor Nunes[4], Maria José Brito[5], Maria João Cardoso [5], Joana Ribeiro[5], Fátima Cardoso [5], Miguel Godinho Ferreira [1,6 ✉] & Rita Fior [1 ✉]

Despite promising preclinical results, average response rates to anti-VEGF therapies, such as bevacizumab, are reduced for most cancers, while incurring in remarkable costs and side effects. Currently, there are no biomarkers available to select patients that can benefit from this therapy. Depending on the individual tumor, anti-VEGF therapies can either block or promote metastasis. In this context, an assay able to predict individual responses prior to treatment, including the impact on metastasis would prove of great value to guide treatment options. Here we show that zebrafish xenografts are able to reveal different responses to bevacizumab in just 4 days, evaluating not only individual tumor responses but also the impact on angiogenesis and micrometastasis. Importantly, we perform proof-of-concept experiments where clinical responses in patients were compared with their matching zebrafish Patient-Derived Xenografts - zAvatars, opening the possibility of using the zebrafish model to screen bevacizumab therapy in a personalized manner.

[1] Champalimaud Centre for the Unknown, Champalimaud Foundation, 1400-038 Lisbon, Portugal. [2] Gastric Unit, Champalimaud Clinical Center, Champalimaud Foundation, 1400-038 Lisbon, Portugal. [3] Hospital Prof. Doutor Fernando Fonseca, Pathological Anatomy Service, 2720-276 Amadora, Portugal. [4] Surgery Unit B, Hospital Prof. Doutor Fernando Fonseca, 2720-276 Amadora, Portugal. [5] Breast Unit, Champalimaud Clinical Center, Champalimaud Foundation, 1400-038 Lisbon, Portugal. [6] Institute for Research on Cancer and Aging of Nice (IRCAN), Université Côte d'Azur, U1081 UMR7284 UNS, 06107 Nice, France. ✉email: miguel.ferreira@neuro.fchampalimaud.org; rita.fior@research.fchampalimaud.org

Angiogenesis is a well-recognized hallmark of cancer, critically involved in tumor growth and metastatic spread[1]. One of the key players in tumor-induced angiogenesis is VEGF-A, which is often found upregulated in many solid tumors. To target VEGF signaling several therapies were developed with the hope of reverting tumor angiogenesis and therefore tumor growth, referred as antiangiogenic therapies.

Bevacizumab is a humanized monoclonal antibody, whose mechanism of action is to target and neutralize three human VEGF-A isoforms[2,3]. The bound form prevents VEGF-A interaction with its receptors and, consequently, impairs signaling and angiogenesis[2,3]. Currently, bevacizumab is used in the clinic as a single agent or combined with antineoplastic drugs for several advanced cancers. These include metastatic colorectal cancer (CRC), metastatic breast cancer, non-small cell lung cancer, ovarian cancer, metastatic renal carcinomas and glioblastomas[3]. Food and Drug Administration approval of bevacizumab for metastatic breast cancer was withdrawn after several studies and a meta-analysis that showed a very limited efficacy in an unselected breast cancer population[4]. European Medicines Agency, on the other hand, kept its approval. The lack of predictive biomarkers of response/resistance does not allow a correct selection of patients, leading the international breast cancer guidelines to consider bevacizumab only in highly selected cases[5].

Overall average response rates for bevacizumab as monotherapy are neither satisfactory nor consistent, ranging from 5.9% (metastatic melanoma) to 9.3% (metastatic breast cancer), but can reach 54.8% in glioblastoma[6–8]. In addition, bevacizumab treatment can be associated with serious side effects, such as cardiovascular complications, bleeding and renal toxicity[4,9,10].

Although VEGF-A signaling is mostly associated to angiogenesis, tumor cells can also express receptors that respond to this pathway, controlling tumor cell proliferation, survival and migration[11]. It has been shown that VEGF-A may act on different steps of the invasion-metastatic cascade[12–15]. Paradoxically, depending on the tumor cells, VEGF-A inhibition can either reduce or induce metastasis. This effect was observed not only in preclinical mouse models but also in patient responses to therapy[11–29].

Tumor-specific heterogeneous and contradicting responses can relate to the pleiotropic effects of VEGF-A and may explain the poor average overall response rates of bevacizumab and other antiangiogenic therapies. Therefore, identification of patients who benefit from these therapies will increase efficacy rates and avoid unnecessary toxicities and healthcare costs. Unfortunately, no clinical biomarker has been identified to date and the promising in vitro tests currently under development, such as organoids, are unable to evaluate the impact on metastasis. Thus, an in vivo model that directly challenges tumor cells to therapy is urgently needed. Currently, mouse Patient-Derived Xenografts (PDX) are the established model and the adult zebrafish xenografts are emerging[30]. However, mouse PDX take months to implant and to evaluate therapy options and therefore are not feasible within the time frame required for a clinical decision.

Recently, we developed zebrafish xenografts to screen therapeutic options in advanced CRC[31]. Also, we and others have shown that zebrafish xenografts can be used to screen for cancer-driven angiogenesis and metastatic potential[31–36]. This is only possible because the majority of cancer-associated human genes are conserved in zebrafish, both in structure and function. The same applies for the signaling pathways that control cell proliferation, migration, death and differentiation[32,35]. Moreover, vascular growth factors and the respective receptors, as well as the signaling pathways that regulate mammalian vascular development, are also well conserved between humans and zebrafish[37]. Such conservation enables the study of different cancer hallmarks, namely on tumor-induced angiogenesis and metastasis.

With the aim of challenging zebrafish xenografts as an in vivo screening platform for bevacizumab responses, we used several representative tumor cell models of triple negative breast cancer (TNBC) and CRC that express VEGF-A. In just 4 days, we were able to evaluate the reported heterogeneity of responses to bevacizumab treatment on angiogenesis and metastasis, along with its direct impact on tumor biology. As a proof-of-principle, we also generated breast cancer and CRC zebrafish PDX (zPDX or zAvatars) and show that we can detect similar phenotypic profiles. Importantly, we provide two case studies where the zAvatars showed resistance to bevacizumab and their corresponding patients progressed under treatment, corroborating our results. Overall, our results suggest that zAvatars constitute a promising in vivo model to decide bevacizumab treatment in a personalized manner.

## Results

**Different tumoral features displayed in zebrafish xenografts.** We started by selecting several cancer cell lines known to express VEGF-A (Supplementary Table 1) and characterized the corresponding xenografts according to several hallmarks of cancer, including cell proliferation, cell death, metastatic and angiogenic potentials. For breast cancer, we selected Hs578T and MDA-MB-468, both considered TNBC representative cell lines[38]. Whereas Hs578T was derived from a primary breast carcinoma, MDA-MB-468 was isolated from a pleural effusion metastasis originated from a primary breast adenocarcinoma. For CRC, we selected HCT116, SW620 and HT29 as representative cell lines. HCT116 was isolated from a primary colorectal carcinoma, HT29 from a primary colorectal adenocarcinoma and SW620 was derived from a lymph node metastasis, whose primary tumor was localized in colon[39]. On Supplementary Table 1, we summarize the expression profile for the relevant cancer-driven mutations, expression of ligands and receptors of VEGF family, as well as sensitivity to bevacizumab, according to the available literature. All tumor cell lines were injected into the perivitelline space (PVS) of 2 days post fertilization (dpf) zebrafish larvae and analyzed at 4 days post injection (dpi) for the several cancer hallmarks (Fig. 1a–e).

TNBC and CRC cell lines revealed different basal levels of apoptosis, with representative TNBC cells showing higher level of activated Caspase3 than CRC cells (Fig. 1f–j, u). In contrast, quantification of proliferation shows that CRC representative tumors present higher proliferative rates (Fig. 1f'–j', v). A balance between diverse cellular events, such as cell proliferation and cell death, can contribute to different tumor sizes as observed at the end of the 4 days assay (Fig. 1f–j, w). Hs578T TNBC xenografts were the smallest tumors (AVG ~281 cells), correlating with their higher basal apoptotic index and low proliferation. In contrast, HCT116 CRC xenografts were the largest tumors (AVG ~2420 cells), which exhibited the highest proliferation rate and lowest apoptotic index (Fig. 1u–w).

Given our goal of testing the antiangiogenic and metastatic impact of bevacizumab treatment in zebrafish xenografts, we analyzed tumor-induced angiogenesis by generating xenografts in Tg(fli1:eGFP) transgenic zebrafish hosts, which have the vasculature labeled with eGFP[40]. Most tumors showed a well-vascularized network at their base (Fig. 1k–o, x). However, Hs578T and HT29 had the additional capacity to recruit a dense vessel network that infiltrates into the tumors, being Hs578T tumors the most vascularized ones (vessel density ~32.4%) (Fig. 1x). In order to assess the metastatic potential of each tumor, we quantified the capacity of tumor cells to colonize the caudal hematopoietic tissue (CHT) located in the tail region, the most

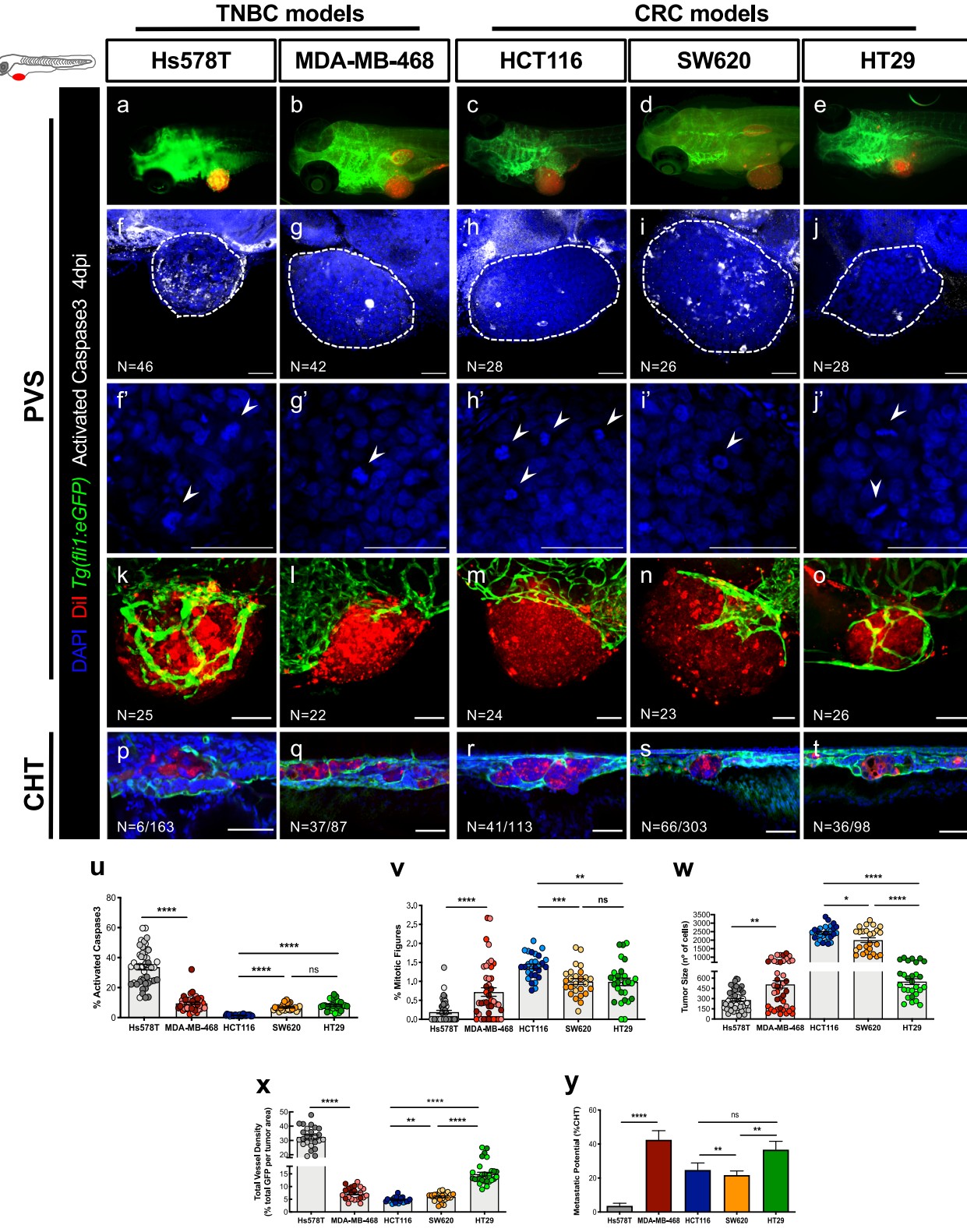

distant site from injection. Interestingly, Hs578T TNBC xenografts that had the most vascularized tumors presented the lowest capacity to colonize the CHT region (only ~3.6% of xenografts presented micrometastasis in the CHT, $N = 6/163$, Fig. 1x, y). In contrast, the MDA-MB-468 TNBC that recruits blood vessels only to the base of the tumor, showed the highest metastatic potential (~42.5%, $N = 37/87$, Fig. 1y). Nevertheless,

HT29 CRC that presented high vessel density (~14.9%, $N = 28$) also showed a high metastatic potential (~36.7%, $N = 98$, Fig. 1x, y). Our results thus show that there is no direct correlation between vessel density and metastatic potential. Consistently, previous studies performed in pancreatic and CRC tumor models were also unable to show a correlation between angiogenesis and metastasis[41,42].

**Fig. 1 Characterization of zebrafish TNBC and CRC xenografts models.** Human cancer cell lines (Hs578T, MDA-MB-468, HCT116, SW620 or HT29) were fluorescently labeled with DiI (red) and injected into the perivitelline space (PVS) of 2 days post fertilization (dpf) *Tg(fli1:eGFP)* zebrafish larvae (**a-e**). At 4 days post injection (dpi), zebrafish xenografts were evaluated regarding: apoptotic index—% of activated Caspase3 (**f-j, u**), mitotic index—% mitotic figures (**f'-j', v**), tumor size (**f-j, w**), angiogenic capacity (**k-o, x**) and metastatic potential (**p-t, y**). White arrowheads indicate mitotic figures. Apoptotic index (**u**, ****$P < 0.0001$), mitotic figures (**v**, Hs578T versus MDA-MB-468 ****$P < 0.0001$, ***$P = 0.0002$, **$P = 0.0049$), tumor size (**w**, ****$P < 0.0001$, **$P = 0.0065$, *$P = 0.0129$), total vessel density (**x**, ****$P < 0.0001$, **$P = 0.0045$) and metastatic potential (**y**, ****$P < 0.0001$, HCT116 versus SW620 **$P = 0.0036$, SW620 versus HT29 **$P = 0.0048$, Fisher's exact test) are expressed as AVG ± SEM. The number of xenografts analyzed are indicated in the representative images and each dot represents one zebrafish xenograft. Results are from 3 (Hs578T and MDA-MB-468) and 2 (HCT116, SW620 and HT29) independent experiments, which are highlighted in different colors corresponding to each individual experiment. Statistical analysis was performed using an unpaired *t*-test or Fisher's exact test. Statistical results: (ns) > 0.05, *$P \leq 0.05$, **$P \leq 0.01$, ***$P \leq 0.001$, ****$P \leq 0.0001$. Scale bars represent 50 µm. All images are anterior to the left, posterior to right, dorsal up and ventral down.

**Differential tumor response to bevacizumab treatment**. To test the effects of bevacizumab, we first tested several concentrations of bevacizumab in the Embryonic medium (E3) (larvae swimming medium), based on maximum plasma concentration (Cmax) found in patients (90–140 µg/mL)[43], which did not induce any mortality (Supplementary Fig. 1a, see details in "Methods"). We chose ~2 times the Cmax (250 µg/mL) as our working concentration, since above this concentration would not be feasible due to lack of antibody availability for animal experiments. To further test the absorption of the bevacizumab antibody, we conjugated the commercial bevacizumab to FITC. Hs578T cell line was used to validate the protocol, since expresses high levels of VEGF-A (Supplementary Table 1) and Hs578T tumors were the most vascularized (Fig. 1k, x). At 3 dpi, when tumors are established and vasculature developed, Hs578T xenografts were incubated with bevacizumab-FITC for 24 h in the E3 and imaged by confocal microscopy (Supplementary Fig. 1b–f'). As depicted in Supplementary Fig. 1c–f', bevacizumab-FITC is detected specifically in the tumor region and not in the tail of the same fish, demonstrating absorption and specific binding to the human tumor cells.

Although bevacizumab is absorbed, we observed that to have a clear phenotype (Fig. 2), not only tumor cells had to be resuspended in bevacizumab (100 ng/mL)[44] prior to injection but also xenografts had to be incubated with bevacizumab in the E3 for 3 consecutive days (250 µg/mL).

After protocol optimization, TNBC and CRC zebrafish xenografts were generated and treated with bevacizumab (Fig. 2a–e', f, g, h). At 4 dpi, we were unable to detect significant reduction of proliferation in any of the represented tumors (Fig. 2f, quantification of mitotic figures or phosphohistone H3, Supplementary Fig. 2a–a", b, c).

However, analysis of cell death revealed a significant induction of apoptosis in Hs578T TNBC xenografts (~1.6 fold, ****$P < 0.0001$) accompanied by a reduction in tumor size (~40% tumor reduction, ****$P < 0.0001$) (Fig. 2g, h). Although we did not observe an induction of apoptosis in other tumors, bevacizumab induced a significant reduction of tumor size in both HCT116 (~22% tumor reduction, ****$P < 0.0001$) and SW620 (~25% tumor reduction, ****$P < 0.0001$) CRC xenografts (Fig. 2g, h). These results may point to an earlier onset of Caspase3 induction or induction of alternative pathways of cell death. Even though MDA-MB-468 and HT29 cells express ligands and receptors for the VEGF family (Supplementary Table 1), both behaved as resistant to bevacizumab and none of the parameters analyzed were altered upon treatment (Fig. 2f–h). These results are consistent with previous studies performed in mouse xenografts[45,46].

**Bevacizumab modulates angiogenesis and vessel normalization**. Next, we evaluated the impact of bevacizumab on the tumor microenvironment, in particular on the angiogenic potential of each tumor model (Fig. 3a–g, compare with Supplementary Fig. 3—non-injected larvae).

Hs578T TNBC tumors that exhibited the highest angiogenic potential were also the most affected by bevacizumab treatment, reducing significantly both vessel density (~20% reduction, **$P = 0.0023$) and vessel infiltration (~32% reduction, **$P = 0.0027$, Fig. 3a–a', f, g, see representative Z stack for vessel infiltration—Supplementary Movies 1 and 2 and Supplementary Fig. 4a–a' for serial slices). Along with Hs578T, HT29 CRC tumors also showed a high angiogenic potential. However, bevacizumab treatment neither affected the overall vessel density or infiltration in HT29 tumors (Fig. 3e–e', f, g, see representative Z stack for vessel infiltration—Supplementary Movies 3, 4 and Supplementary Fig. 4b–b' for serial slices). These results are consistent with previous studies performed in HT29 mouse xenografts where anti-VEGF treatment had no antiangiogenic effect[45]. In the remaining tumor models, MDA-MB-468, HCT116 and SW620 with simple vascularized periphery, we did not observe a significant impact on vessel network (Fig. 3b–d', f), suggesting that vessels recruited to the base of these tumors are dependent on other pro-angiogenic factors. Moreover, our results also showed that the most vascularized tumors were also smaller in size, suggesting that, in this short assay, angiogenesis is not a consequence of tumor hypoxia but, rather, a read-out of the balance of the available pro- and anti-angiogenic factors released by tumor cells and, therefore, a useful reporter for angiogenic potential (Supplementary Fig. 5).

Since bevacizumab is administered in the clinic intravenously, we next investigated the impact of injecting bevacizumab in circulation (100 ng/mL) in the most sensitive xenograft Hs578T. We designed two settings: (i) injecting daily since 1 hpi during 4 days (Supplementary Fig. 6a) and (ii) injecting later when tumor vessels are already formed (3 dpi), also during 4 days (Supplementary Fig. 6h). In both settings, bevacizumab is able to induce its antitumor effect. In setting a, bevacizumab induced a significant increase of apoptosis (*$P = 0.01$) without tumor size reduction, accompanied by a reduction of vessel infiltration (*$P = 0.016$) (Supplementary Fig. 6b–g). In setting h, we performed a time course, monitoring daily each xenograft. At the end of the assay we observed a clearance of ~30% of tumors in bevacizumab-treated xenografts but no clearance in the controls. Also, bevacizumab-treated xenografts seem to present a higher incidence of tumor size reduction, but we could not detect reduction of angiogenesis (Supplementary Fig. 6i–o). Our original protocol (cells preincubated with bevacizumab + bevacizumab in E3) provides a more robust effect. This may be due to the dosage (in the E3 we apply a much higher concentration), but also because of the timings. In our original protocol, cells have bevacizumab exerting its activity as soon as cells are prepared. Thus, if the assay is extended one more day or if we increase the dosage and maintain the timings, we would probably observe the tumor size reduction phenotype. In the second setting, the assay

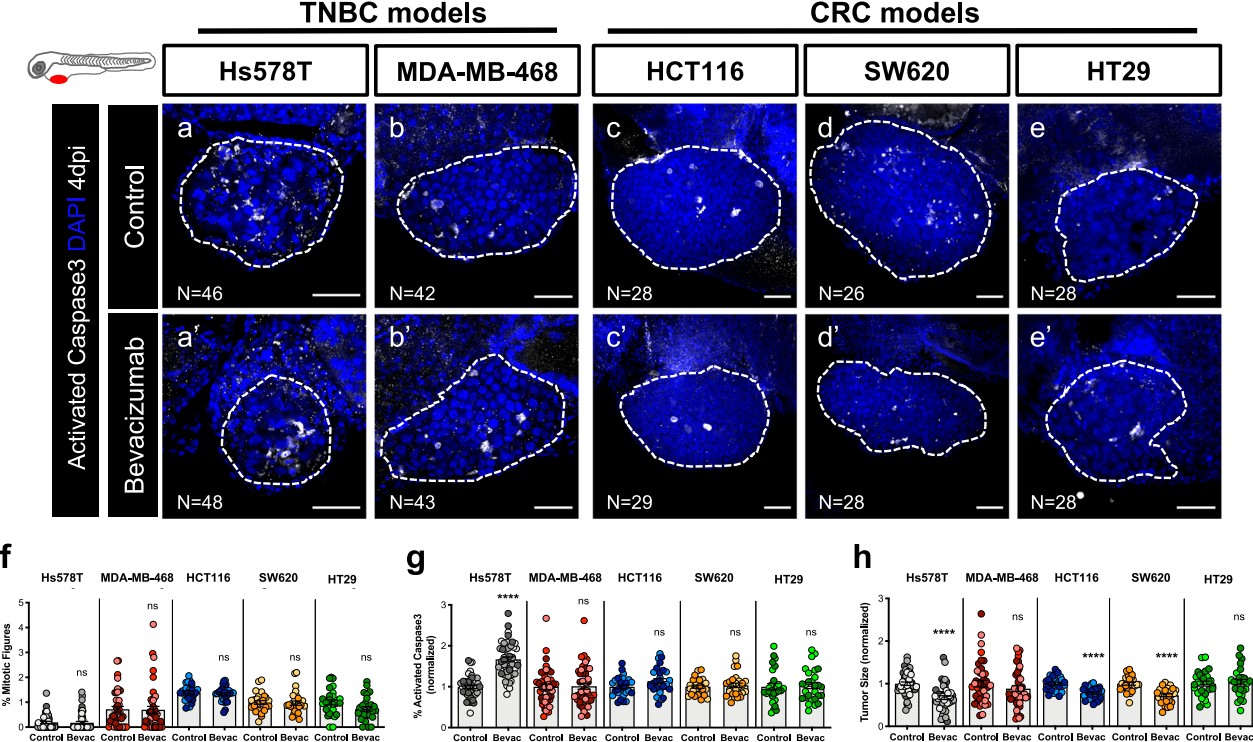

**Fig. 2 Zebrafish xenografts reveal different tumor responses to bevacizumab.** Human cancer cell lines (Hs578T, MDA-MB-468, HCT116, SW620 or HT29) were injected into the PVS of 2 dpf *Tg(fli1:eGFP)* zebrafish larvae. Zebrafish xenografts were treated in vivo with bevacizumab and compared with untreated controls. At 4 dpi, zebrafish xenografts were imaged by confocal microscopy (**a–e′**). The percentage of mitotic figures (**f**), apoptosis (**g**, ****$P <$ 0.0001) and tumor size (**h**, ****$P <$ 0.0001) were quantified. The outcomes are expressed as AVG ± SEM. The number of xenografts analyzed are indicated in the representative images and each dot represents one zebrafish xenograft. Results are from 3 (Hs578T and MDA-MB-468) and 2 (HCT116, SW620 and HT29) independent experiments, which are highlighted in different colors corresponding to each individual experiment. Statistical analysis was performed using an unpaired *t*-test. Statistical results: (ns) > 0.05, *$P \leq$ 0.05, **$P \leq$ 0.01, ***$P \leq$ 0.001, ****$P \leq$ 0.0001. Scale bars represent 50 μm. All images are anterior to the left, posterior to right, dorsal up and ventral down.

terminates later in development (9 dpf) raising the possibility of some interplay with the immune system that can lead to tumor clearance. In summary, although intravenously experiments mirror an ideal experimental design, they are time-consuming and demanding to implement. Also, many xenografts die during procedure, making it very difficult to obtain high numbers, compromising the robustness of the assay. Therefore, we would argue that our initial protocol is a good compromise of practicality and feasibility.

Thus, we returned to our original protocol to further characterize the tumor-related vessels in the most sensitive and angiogenic tumors. To this end, a filament analysis was performed, revealing that bevacizumab leads to a reduction of the number of branching points, but has no impact on overall vessel length in Hs578T tumors (Fig. 3h–j', k, l).

It has been shown that, in general, tumor-related vessels are heterogenous and tortuous and poorly perfused, consequently not fully functional[47]. However, antiangiogenic therapies, by rebalancing the pro- and anti-angiogenic factors, may revert this phenotype leading to a vascular normalization[47]. Therefore, to investigate vessel functionality and the impact of bevacizumab on vessel normalization, we injected Hs578T cancer cells into *Tg(fli1: eGFP; Tg(gata1:DsRed)*[48], in which erythrocytes are labeled with DsRed and the vascular system with eGFP (Fig. 3). In this way, we could evaluate if the tumor-related blood vessels were functional, i.e., able to transport red blood cells. Hs578T untreated xenografts showed vascularized tumors, but we could not find any erythrocytes inside these very abnormal and tortuous vessels. In

contrast, upon bevacizumab treatment, we could observe a clear increase in the number of tumors with red blood cells inside the vessels, suggesting that bevacizumab may normalize vessel function (Fig. 3m–n', o, $P =$ 0.073).

Altogether, our data suggest that bevacizumab can modulate not only vessel recruitment, but also vessel functionality.

**Bevacizumab can enhance or impair micrometastatic potential.** VEGF-A is a pleiotropic molecule that can also impact on different steps of the invasion-metastatic cascade[12–15]. Previous studies report that antiangiogenic therapies may have contradictory effects, in some cases reduce metastatic disease while in others promote it[11–29]. Therefore, we tested whether we could detect these variable responses in our assay. We first evaluated the impact of bevacizumab on the capacity of tumor cells to disseminate and colonize distant sites, in particular, the CHT region. At 4 dpi, the presence of micrometastasis in the CHT region was quantified (Fig. 4a). Control Hs578T TNBC tumors had a reduced capacity to form micrometastasis in the CHT region, as previously observed (Fig. 1y). However, upon bevacizumab treatment, we observed a 3.5-fold increase in the frequency of micrometastasis in the CHT (from ~3.6% to ~12.8%, **$P =$ 0.0041, Fig. 4b). In clear contrast, in MDA-MB-468 TNBC tumors that exhibited the highest metastatic potential, bevacizumab treatment reduced the frequency of micrometastasis to half (from ~42.5% to ~19.7%, **$P =$ 0.0017, Fig. 4b). For CRC tumors, bevacizumab increased the incidence of CHT metastasis

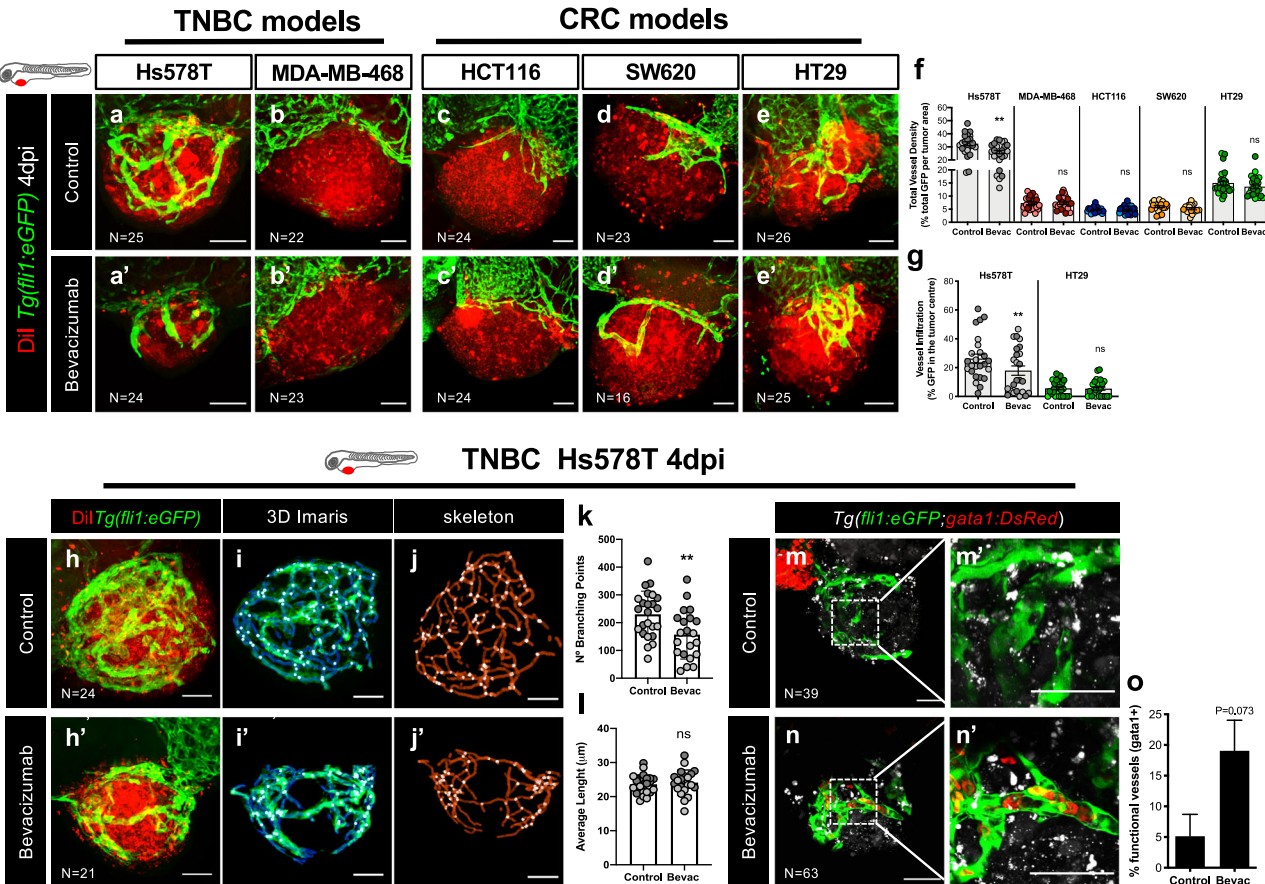

**Fig. 3 Bevacizumab reduces angiogenesis and promotes vessel normalization.** Human cancer cell lines (Hs578T, MDA-MB-468, HCT116, SW620 or HT29) were fluorescently labeled with DiI (in red) and injected into the PVS of 2 dpf *Tg(fli1:eGFP)* zebrafish larvae. Zebrafish xenografts were treated in vivo with bevacizumab and compared with untreated controls. At 4 dpi, zebrafish xenografts vasculature was imaged by confocal microscopy (max Z-projections) (**a–e′**). Total vessel density (**f**, **\*\*P = 0.0023**) and vessel infiltration (**g**, **\*\*P = 0.0027**) were quantified. Filament analysis was performed in Hs578T tumor-related vasculature. Hs578T xenografts confocal images of untreated and Bevacizumab-treated (**h**, **h′**) were used to perform 3D projections using Imaris (**i–i′**) and skeletonized images on ImageJ (**j–j′**). Number of branching points (**k**, **\*\*P = 0.007**) and average vessel length (**l**, **P = 0.61**) were quantified by filament analysis. To analyze the functionality of Hs578T tumor-related vessels, tumor cells were labeled with DeepRed (in gray) to generate xenografts in *Tg(fli1:eGFP; gata1:DsRed)* zebrafish larvae. Xenografts were treated in vivo with bevacizumab and compared with untreated controls. At 4 dpi, zebrafish xenografts were mounted in low melting agarose to be visualized by live-imaging confocal microscopy (**m–n′**). The percentage of xenografts with erythrocytes inside blood vessels was quantified (**o**). All outcomes are expressed as AVG ± SEM. The number of xenografts analyzed are indicated in the representative images and each dot represents one zebrafish xenograft. Results are from 2 independent experiments, which are highlighted in different colors corresponding to each individual experiment. Statistical analysis was performed using an unpaired *t*-test or Fisher's exact test. Statistical results: (ns) > 0.05, \*P ≤ 0.05, \*\*P ≤ 0.01, \*\*\*P ≤ 0.001, \*\*\*\*P ≤ 0.0001. Scale bars represent 50 μm. All images are anterior to the left, posterior to right, dorsal up and ventral down.

in SW620 xenografts (from ~22% to ~31%, \*P = 0.0112, Fig. 4b) but the impact of treatment on HCT116 and HT29 CRC xenografts was not significantly altered (P = 0.88 and P = 0.88, respectively, Fig. 4b).

To further confirm these results, we investigated the presence of micrometastasis in other organs beyond the CHT, namely the brain, eye and gills and quantified the incidence of micrometastasis in these multiple organs (Supplementary Fig. 7a–e). Strikingly, our results show that bevacizumab was able to significantly reduce the incidence of micrometastasis in multiple sites per animal in MDA-MB-468 (from ~43.7 to ~23.4%, \*\*P = 0.0081), HCT116 (from ~19 to ~2.6%, \*P = 0.03) and HT29 (from ~44 to ~20%, \*\*P = 0.0061) (Fig. 4c—presence of micrometastasis in two or more sites). In contrast, in Hs578T and SW620 xenografts that showed an increase in the CHT incidence, we also did not observe an increase of metastasis in other sites. These results highlight the complexity of the metastatic process in response to bevacizumab, revealing that

bevacizumab can either promote or decrease the capacity of tumor cells to disseminate and colonize distant sites. In addition, bevacizumab may also influence organ preference, independent of the origin of the tumors (colon or breast) (see Supplementary Fig. 7a–e).

**Bevacizumab impacts on different steps of the metastatic cascade.** Metastasis encompasses a multistep cascade of events that could be divided into two stages[49,50]. The first steps involve tumor cell detachment from the primary tumor by disruption of cell–cell junctions and extracellular matrix degradation, migration plus invasion of adjacent tissues and intravasation into the bloodstream. The later steps comprise tumor cell survival in circulation, extravasation from the blood or lymphatic vessels, and colonization at secondary sites[49,50]. To better understand the effect of bevacizumab on the invasion-metastasis cascade, we used our previously designed assay[31] to

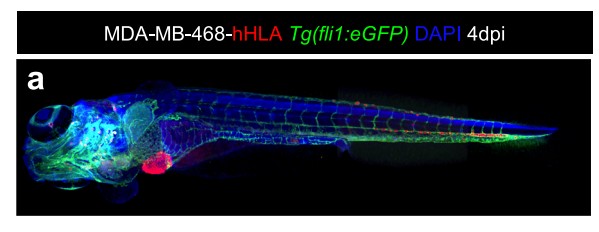

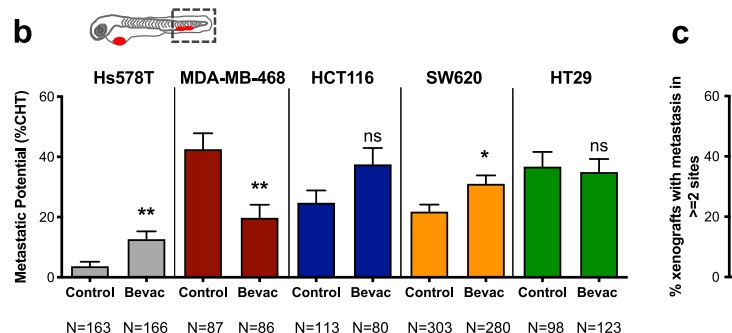

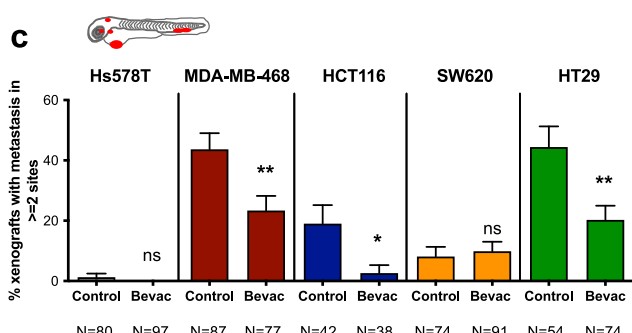

**Fig. 4 Bevacizumab can both promote and impair the metastatic potential of human cancers.** Representative image of an MDA-MB-468 zebrafish xenograft with a tumor in the PVS and several micrometastasis spread throughout the zebrafish larvae body, namely in brain, eye, gills and CHT (**a**). Human cancer cell lines (Hs578T, MDA-MB-468, HCT116, SW620 or HT29) were injected into the PVS of 2 dpf *Tg(fli1:eGFP)* zebrafish larvae. Zebrafish xenografts were treated in vivo with bevacizumab and compared with untreated controls. At 4 dpi, zebrafish xenografts were imaged by a fluorescence stereoscope to detect tumor cells throughout the zebrafish body (**b**). The percentage of xenografts that display micrometastasis in the CHT region was quantified (**b**, Hs578T **$P = 0.0042$, MDA-MB-468 **$P = 0.0017$, SW620 *$P = 0.0112$) and the outcomes are expressed as AVG ± SEM. Results are from 5 (Hs578T and SW620), 3 (MDA-MB-468) and 2 (HCT116 and HT29) independent experiments. The presence of micrometastasis in other organs besides the CHT was also quantified, namely in the brain, eye and gills, in untreated and bevacizumab-treated xenografts and the incidence of micrometastasis in two or more metastatic sites was determined (**c**, MDA-MB-468 from ~43.7 to ~23.4%, **$P = 0.0081$; HCT116 from ~19 to ~2.6%, *$P = 0.0308$; and HT29 from ~44 to ~20%, **$P = 0.0061$). Statistical analysis was performed using a Fisher's exact test. Statistical results: (ns) > 0.05, *$P \leq 0.05$, **$P \leq 0.01$, ***$P \leq 0.001$, ****$P \leq 0.0001$.

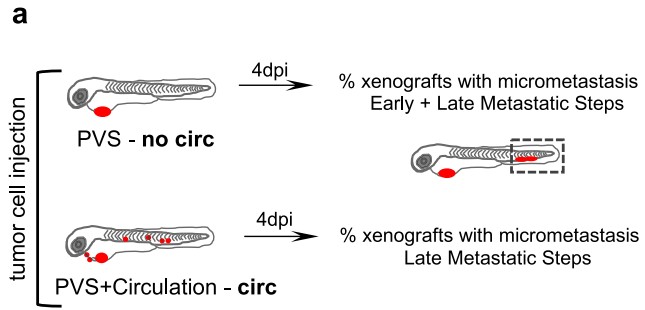

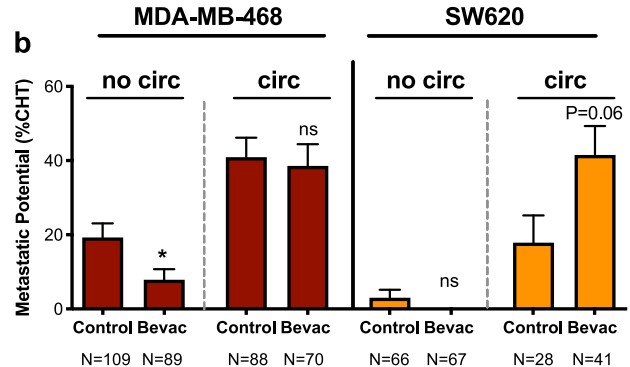

**Fig. 5 Bevacizumab can modulate early and late metastatic steps.** Schematic representation to distinguish between early and late metastatic steps (**a**). Injected xenografts were sorted into two groups—xenografts with only cells in the PVS and xenografts with cells in circulation. MDA-MB-468 and SW620 micrometastasis in the CHT region were quantified from both groups (**b**, MDA-MB-468 *$P = 0.0246$ and SW620 $P = 0.06$). Results are from 2 independent experiments and outcomes are expressed as AVG ± SEM. The number of xenografts analyzed are indicated below the graphs. Statistical analysis was performed using a Fisher's exact test. Statistical results: (ns) > 0.05, *$P \leq 0.05$, **$P \leq 0.01$, ***$P \leq 0.001$, ****$P \leq 0.0001$.

distinguish between these early and late events and address the potential/proficiency of the different tumor cells to perform these steps (Fig. 5a). MDA-MB-468 and SW620 tumors, which displayed opposite phenotypes to bevacizumab regarding metastatic potential, were injected either in the PVS alone (i.e., without cells in circulation—no circ) or directly into circulation (circ) and PVS, thus bypassing the first metastatic steps. At 4 dpi, we quantified frequency of xenografts with the presence of micrometastasis in the CHT (Fig. 5b). Strikingly, our data suggest that bevacizumab can impact both in early and late stages of the invasion-metastasis cascade. Upon bevacizumab

treatment, we observed a ~60% reduction of the capacity of MDA-MB-468 TNBC tumors to go through the early steps (from ~19.27% to ~7.8%, *$P = 0.02$, Fisher's exact test, Fig. 5b). In contrast, when cells were directly injected into circulation, bevacizumab treatment showed a tendency to double the incidence of SW620 CRC micrometastasis (from ~19% to ~44%, $P = 0.06$, Fisher's exact test, Fig. 5b). This result suggests that bevacizumab might increase the capacity of SW620 cells to perform the later steps of metastasis formation.

Altogether, our results reveal the possible multiple functions of VEGF-A signaling in the metastatic cascade in a

tumor-dependent fashion: in some cases promoting early metastatic events, whereas in others the later steps.

**Differential response to bevacizumab in zAvatars.** Next, as a proof-of-concept, we tested whether we could observe these paradoxical effects of bevacizumab in zAvatars. We processed three breast cancer and three CRC surgical resected samples (Supplementary Table 2) and treated the resulting zAvatars with bevacizumab. For all we analyzed angiogenesis, induction of cell death by apoptosis, tumor mass and incidence of micrometastasis. In contrast to other surgical resected samples[31], we could not detect any significant recruitment of blood vessels to the tumor mass in these particular zAvatars and, therefore we did not proceed to quantify angiogenesis (Supplementary Fig. 8a–f').

Nevertheless, we could observe a variety of phenotypes, some reminiscent of the paradoxical effects observed in the tumor cell lines models, but also new profiles (Fig. 6a–c). Due to the reduced amount of sample, our zPDX numbers were low, reducing the statistical power of analysis. In order to address this, we performed an additional effect size analysis—Cohen' D with a Hedges' $g$ correction for low number of samples (xenografts, see "Methods"). Nonetheless, we are aware that we cannot take major conclusions, only profiles.

zPDX#1 presented a phenotype reminiscent of SW620, with bevacizumab showing a tendency to shrink the tumor size (Fig. 6b, ~50% reduction, $P = 0.095$, $g = 1.11$) but an increase in micrometastasis incidence (Fig. 6c, from ~10% to ~45%, $P = 0.087$, $g = 0.78$, see Supplementary Fig. 8g–l). On the other hand, zPDX#2 and zPDX#4 had a profile reminiscent to MDA-MB-468, with bevacizumab having no antitumor effect (Fig. 6b, $P = 1.0$ and $g = 0.03$, $P = 0.42$ and $g = 1.03$, respectively), but a tendency to reduce the micrometastasis incidence (Fig. 6c, zPDX#2 from ~42% to 0%, $P = 0.072$, $g = 1.06$; zPDX#4 from ~20% to 0%, $P = 0.167$, $g = 0.64$). zPDX#3 exhibited a profile similar to Hs578T TNBC, where we could detect a significant increase in apoptosis upon bevacizumab treatment (Fig. 6a, from ~6.5% to ~18.7%, $*P = 0.01$, $g = 1.75$), accompanied by a tendency to shrink the tumor (Fig. 6b, ~21% reduction, $P = 0.42$, $g = 0.56$) but increase micrometastasis frequency (Fig. 6c, from ~12% to ~53%, $P = 0.26$, $g = 0.44$).

In contrast, zPDX#5 and zPDX#6 showed a different profile. In zPDX#5 we observed an increase in apoptosis (Fig. 6a, from ~22% to 42%, $*P = 0.01$, $g = 1.51$) and in zPDX#6 a shrinkage of the tumor mass (Fig. 6b, ~44% reduction, $*P = 0.03$, $g = 1.08$), both accompanied by a tendency to reduce micrometastasis incidence (Fig. 6c, zPDX#5 from ~32% to ~11%, $P = 0.08$, $g = 0.49$; zPDX#6 from ~62% to 37%, $P = 0.07$, $g = 0.50$). In these last "case studies," bevacizumab not only would control the growth of localized tumors but also possibly prevent the development of occult micrometastasis at distant sites.

**Comparison of patient's response with their matching zAvatars.** Next, in order to compare responses to bevacizumab between the zebrafish model and the corresponding patient clinical response, we searched in the Champalimaud Foundation cryopreserved biobank for patients that during the course of their disease might have been treated with bevacizumab. Within this biobank, we were able to retrieve two patients who had available sample material and could constitute two case studies to generate zAvatars.

The first case is a 60-year-old male patient, herein Patient#7, that was diagnosed with a colon adenocarcinoma in April 2017 with synchronous peritoneal carcinomatosis and treated with CAPOX (full clinical history in Supplementary Fig. 9a, see legend). The tumor progressed under CAPOX treatment and was subsequently submitted to surgery (October 2017). Pathology of the surgical specimen revealed a high-grade mucinous adenocarcinoma. In late January 2018 postoperative chemotherapy was proposed with CAPOX + bevacizumab. After three cycles, whole body computerized tomography scan imaging revealed "de novo" parenchymal and subcapsular liver lesions, as well as mesenteric nodules and abdominal wall lesions not previously present (Fig. 7a–c'). Clinical and imaging progression was assumed. After three subsequent lines of treatment (FOLFIRI + bevacizumab, TAS-102 + bevacizumab, Regorafenib) to which the disease was refractory, the patient died. We had access to a surgical specimen from Patient#7 before bevacizumab treatment. We processed this sample (Supplementary Fig. 10a–a' and Supplementary Table 3) and generated the matching zPDX for bevacizumab testing. We analyzed apoptosis and metastatic potential and observed no significant antitumor effect in either parameter (Fig. 7d–e', f, g, induction of apoptosis $P = 0.47$, $g = 0.29$; metastatic potential from ~10% to ~14%, $P = 0.73$ Fisher's exact test, $g = 0.11$). Thus, consistent with the patient refractory response the corresponding zAvatars were not responsive to bevacizumab treatment.

The second patient was a 64-year-old man with a long clinical history, who underwent surgery for a left colon adenocarcinoma in 2011 followed by several rounds of treatment (full clinical history in Supplementary Fig. 9b, see legend). The biopsy that we had access was retrieved after bevacizumab treatment resistance (i.e., progression under treatment and "de novo" lesions, Fig. 7h–j'). Although this case does not constitute the ideal setting for our study, we sought to investigate whether we could detect resistance in the matching zAvatars. The biopsy was processed (Supplementary Fig. 10b–b' and Supplementary Table 3) and zPDX were generated and treated with bevacizumab. Our results show that bevacizumab had no significant impact on induction of apoptosis in the tumor (Fig. 7m $P = 0.18$, $g = 0.82$). However, and although it did not reach statistical significance, we could detect an increment in the incidence of micrometastasis in the CHT when treated with bevacizumab (Fig. 7n from ~8% to ~23%, ~2.8 times more micrometastasis, $P = 0.11$ Fisher's exact test, $g = 0.42$). These results suggest that resistance to bevacizumab treatment in the zPDX may reflect the patients' outcome. Overall, our results show the feasibility of the zPDX assay to inquire the outcome of bevacizumab treatment in patients.

## Discussion

Anti-VEGF-A or antiangiogenic therapies have been under intensive debate, especially due to the controversial effects of these therapies on metastasis and disease progression. Although these therapies can control established primary tumor and metastasis in some patients, later, after treatment withdrawal or even under treatment, some tumors progress[51]. This progression may be manifested by regrowth of the established disease or the appearance of additional metastatic lesions[51]. It has been suggested that this progression may be the result of multiple mechanisms of resistance/adaptation that occur after months of treatment[4,16]. However, it is also under debate whether these antiangiogenic therapies, which control established disease, could simultaneously promote "de novo" invasive and micrometastatic lesions[51]. In other words, these treatments may potentiate the development of hidden micrometastasis at distant sites that will only manifest after a considerable time delay. In contrast, there are also many reports that suggest that antiangiogenic therapies

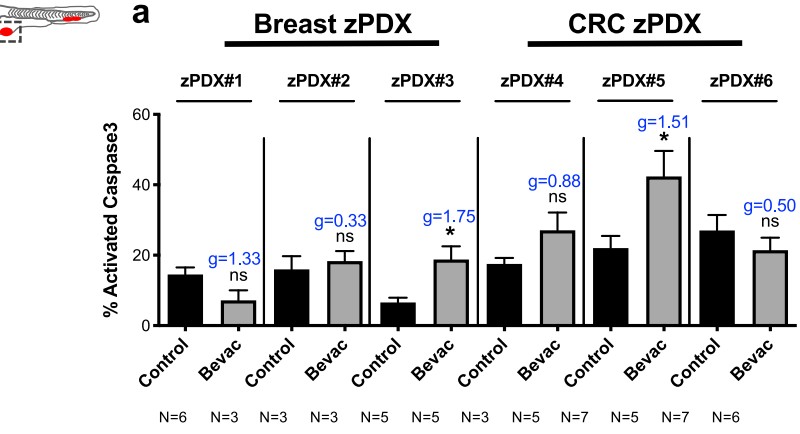

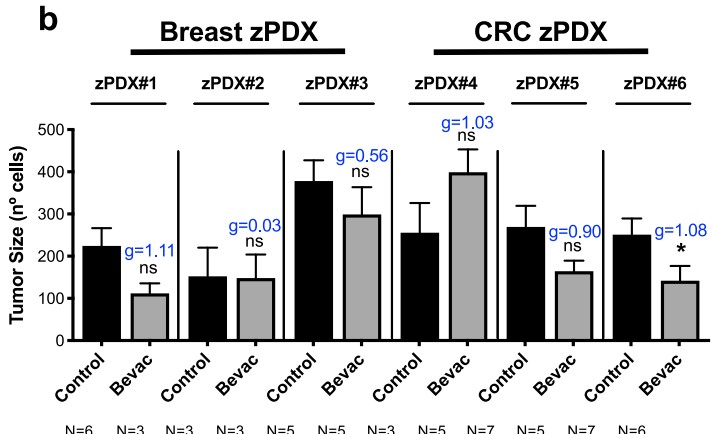

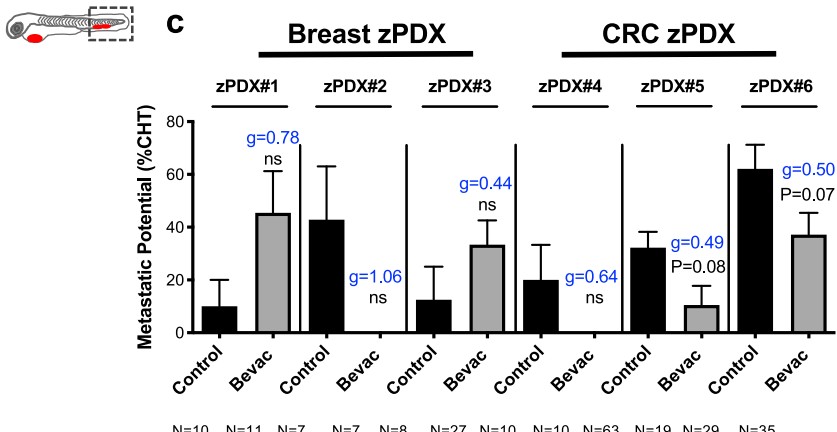

**Fig. 6 zPDX reveal different response profiles to bevacizumab.** Human breast cancer or CRC surgical resected samples were injected into the PVS of 2 dpf *Tg(fli1:eGFP)* zebrafish larvae. zPDXs were treated in vivo with bevacizumab and compared with untreated controls. At 4 dpi, zebrafish xenografts were imaged by confocal microscopy. The percentage of apoptosis (**a**, zPDX#3 *$P = 0.0159$, $g = 1.75$ and zPDX#5 *$P = 0.0101$, $g = 1.51$) and tumor size (**b**, *$P = 0.0381$, $g = 1.08$) were quantified. In parallel, zebrafish xenografts were analyzed in a fluorescent stereoscope to detect micrometastasis in the CHT region, followed by confocal confirmation (**c**). The outcomes are expressed as AVG ± SEM. The number of xenografts analyzed are indicated below the graphs. Results are from 1 independent experiment. Statistical analysis was performed using an unpaired *t*-test for apoptosis and tumor size and a Fisher's exact test for micrometastasis. Statistical results: (ns) > 0.05, *$P \leq 0.05$, **$P \leq 0.01$, ***$P \leq 0.001$, ****$P \leq 0.0001$. Cohen's D 1988 scale of effect size with Hedges' *g* correction (*g*): $g = 0,2$ low; $g = 0.5$ moderate; $g = 0.8$ high.

can reduce metastatic potential[15,27] and therefore constitute a very important antimetastatic therapy. However, no biomarker to identify which patients will benefit from these therapies has been found to date.

Therefore, taking into account the diversity of mechanisms that can be involved in the metastatic process, a functional assay that "reads" the metastatic potential of each tumor in an in vivo

context, independent of the genetic makeup/mechanism, would be fundamental to screen for these therapies with possible detrimental effects in a preclinical or clinical setting.

Here, we show that zebrafish cancer xenografts can reflect the panoply of pro- and anti-metastatic effects of bevacizumab in a time window of just 4 days. In the mice model, it has been shown that antiangiogenic therapy can increase the incidence of

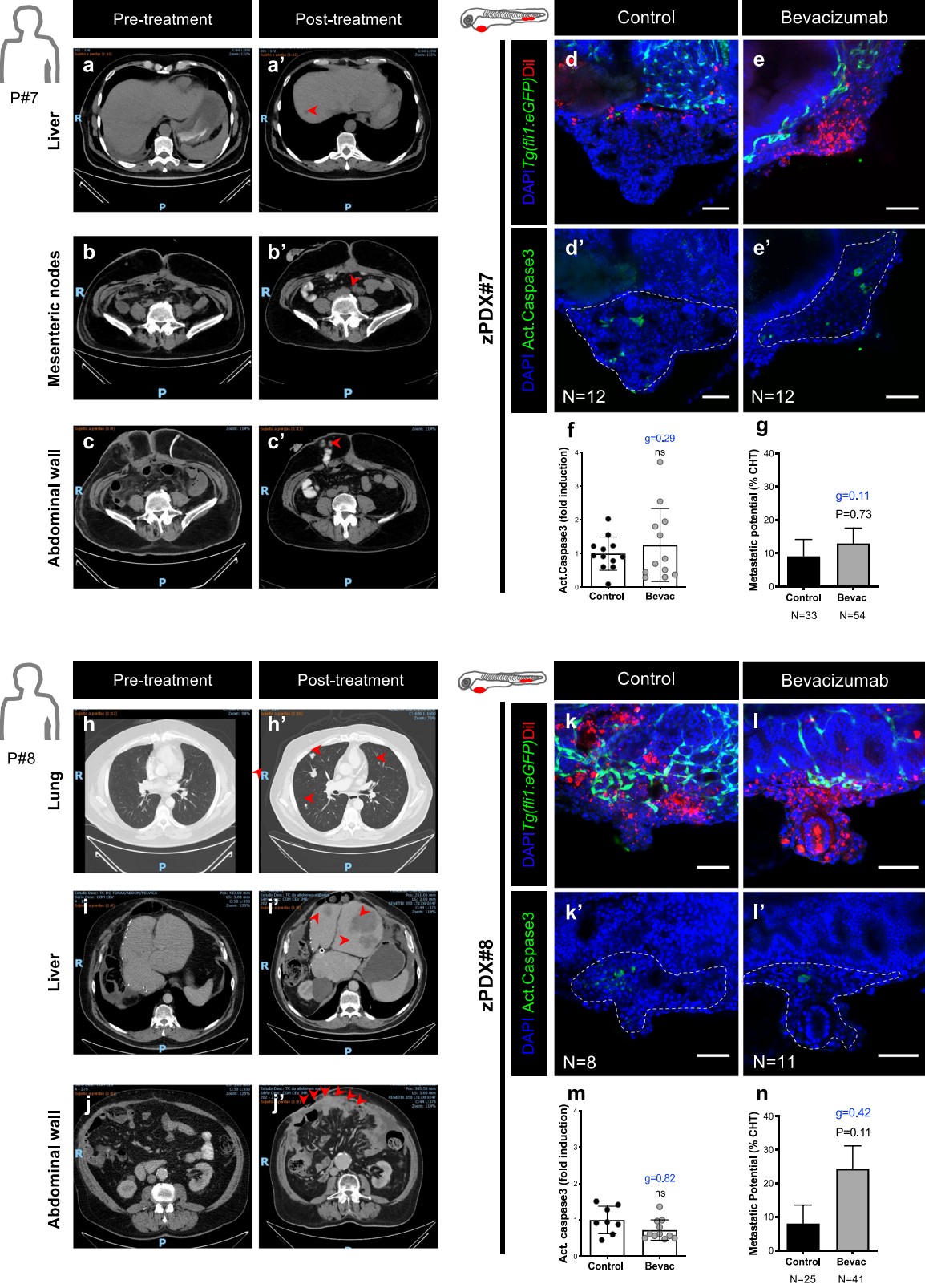

COMMUNICATIONS BIOLOGY | (2020)3:299 | https://doi.org/10.1038/s42003-020-1015-0 | www.nature.com/commsbio

metastasis in a time frame of just 1 week[21,22]. This quick impact of 1 week in mice and 4 days in zebrafish suggest that late metastatic progression in patients may not be a sole consequence of a slow process of resistance/adaptation, but rather as a result of a micrometastatic potential already present in the beginning of therapy, which can be either enhanced or reduced by treatment. The zebrafish model provides a reduction of scale to single cell

resolution, enabling us to detect single circulating cells and micrometastasis composed of just a few cells. Therefore, this reduction of size allows also a reduction of the time scale, enabling us to evaluate the early effects of therapy on escaping/circulating tumor cells, which in patients in early stages are undetectable, but in the future might give rise to metastasis. Finally, we show correlation of resistance in zAvatars with clinical

**Fig. 7 zPDX bevacizumab treatment response may predict relapse and correlates with patients' outcome.** Computerized tomography scans of Patient#7 (**a–c'**) and Patient#8 (**h–j'**) of different regions, including liver (**a**, **a'**, **i**, **i'**), lung (**h**, **h'**), mesenteric region (**b**, **b'**) and abdominal wall (**c**, **c'**, **j**, **j'**) in both pre- and post-treatment settings. Red arrows highlight metastasis. Human patient surgical/biopsy samples were injected into the PVS of 2 dpf *Tg(fli1:eGFP)* zebrafish larvae. zPDXs were treated in vivo with bevacizumab and compared with untreated controls. At 4 dpi, zebrafish xenografts were imaged by confocal microscopy (**d–e'**, **k–l'**). Activated caspase3 was quantified in both groups (**f**, zPDX#7 $P = 0.4682$ and $g = 0.29$; **m**, zPDX#8 $P = 0.1817$ and $g = 0.82$). In parallel, the percentage of xenografts that display micrometastasis in the CHT region was quantified (**g**, zPDX#7 $P = 0.73$ and $g = 0.11$; **n**, zPDX#8 $P = 0.11$ and $g = 0.42$). The outcomes are expressed as AVG ± SEM. Number of zPDX analyzed for each condition is indicated in the figure. Results are from 1 independent experiment. Statistical analysis was performed using an unpaired *t*-test for apoptosis and a Fisher's exact test for micrometastasis. Statistical results: (ns) > 0.05, *$P \leq 0.05$, **$P \leq 0.01$, ***$P \leq 0.001$, ****$P \leq 0.0001$. Cohen's D 1988 scale of effect size with Hedges' *g* correction (*g*): $g = 0{,}2$ low; $g = 0.5$ moderate; $g = 0.8$ high. Scale bars represent 50 µm. All images are anterior to the left, posterior to right, dorsal up, and ventral down.

resistance/progression under bevacizumab treatment in two case studies.

Overall, our results suggest that zAvatars are a promising model to screen bevacizumab treatment in a personalized manner to avoid unwanted effects and healthcare costs. The zAvatar rapid assay will not be able to predict the result of slow resistance/adaptation mechanisms, but may predict which patients will have more chance to have a positive/negative immediate response and possibly predict the development of future micrometastasis. Future work will be aimed at testing the predictive value of the zAvatar model to forecast the effects of bevacizumab on disease progression with many more patients.

## Methods

**Animal care and handling.** In vivo experiments were performed in the zebrafish model (*Danio rerio*), which was maintained and handled in accordance with European Animal Welfare Legislation and Champalimaud Fish Platform Program. The experiments of this project were performed at the Champalimaud Vivarium, which is licensed by the National Authority for Animal Health (DGAV) and complies with European Guidelines (2010/63/EU), National Laws (113/2013) and FELASA guidelines and recommendations concerning laboratory animal welfare, scientific use and proper education/training of all personnel performing animal work. The study and procedures were approved by the Champalimaud Animal Welfare & Ethical Review Body (ORBEA) and DGAV.

**Zebrafish lines.** *Tg(fli1:eGFP)*[40] has a eGFP under the *fli1* promoter expressed specifically in endothelial cells, allowing the visualization of both blood and lymphatic vascular systems. *Tg(gata1:DsRed)*[48] allows visualization of erythrocytes. Transparent zebrafish line, casper[52], was also used for Supplementary Fig. 1.

**Human tissue.** Human breast cancer and CRC resected samples used for zPDXs establishment were obtained from Champalimaud Hospital and Hospital Professor Doutor Fernando Fonseca, respectively, with written informed consents, approved by both Hospital's Ethics Committees. The study was approved by the Ethics Committee of the Champalimaud Foundation and the Hospital Professor Fernando Fonseca. All tumor samples were collected after signed informed consent.

**Human cancer cell lines.** Hs578T and MDA-MB-468 were kindly provided by Mónica Bettencourt Dias' Laboratory at Instituto Gulbenkian Ciência. SW620 and HT29 were originally from American Type Culture Collection. HCT116 cells were provided by Dr Ângela Relógio (Charité Medical University of Berlin). All cell lines were authenticated through short tandem repeat profile analysis and tested for mycoplasma.

**Cell culture.** Cancer cell lines were cultured and expanded to 70–80% confluence in Dulbecco's Modified Eagle Medium High Glucose (Biowest) supplemented with 10% fetal bovine serum (Sigma-Aldrich) and 1% Penicillin-Streptomycin 10,000 U/mL (Hyclone). For Hs578T, culture medium was supplemented with insulin at 10 µg/mL (Sigma-Aldrich).

**Cell labeling.** Cells were labeled with lipophilic dyes: vybrant CM-DiI (Vybrant™ CM-DiI Thermo Fisher Scientific) at a concentration of 2 µL/ml for TNBC cells or 4 µL/mL for CRC cell lines or Deep Red (CellTracker™-Thermo Fisher Scientific) at a concentration of 1 µl/ml according to manufacturer's instructions. TNBC cells were resuspended to a final concentration of $0.50 \times 10^6$ cells/µL and CRC cells to $0.25 \times 10^6$ cells/µL.

**Human tissue processing.** The samples were collected in rich media containing a mixture of antibiotics and antifungals and cryopreserved until injection, as previously described[31]. When defrosted, tumor tissues were minced in Mix1 (Supplementary Table 3). Subsequently, CRC tissue was digested with Liberase (Roche) and breast tissue with Collagenase (20 mg/mL, Worthington) plus Hyaluridase (3 mg/mL, Sigma) at 37 °C. Tumor cell suspension was filtered and centrifuged at 1200 rpm for 4 min. For cell labeling, tumor cells were incubated with the fluorescent cell tracker DiI (10 µL/mL) in Mix2 (Supplementary Table 3) for 15 min at 37 °C and then for 5 min on ice. Tumor cells were checked for viability with trypan blue dye exclusion. Cancer cells were resuspended in Mix1 with human EGF (50 ng/mL, Peprotech) to a final concentration of ~$0.25 \times 10^6$ cells/µL. A small aliquot of the processed/dissociated tumor sample was stained with MGG Grunwald-Giemsa (Bio-Optica) method according to the manufacturer's instructions.

**Zebrafish xenografts injection.** Cancer cells were microinjected into the PVS of anesthetized 2 dpf zebrafish. In general, ~800 (bigger cells) to ~1500 (smaller cells) cells are injected, depending on the size of the tumor cells. Zebrafish xenografts were kept at 34 °C until the end of the experiments. At 1 dpi, zebrafish xenografts were screened regarding the presence or absence of a tumoral mass. Xenografts with cells in the yolk sac, cell debris or non-injected zebrafish larvae were discarded, whereas successful ones were grouped according to tumor size. All xenografts were kept at 34 °C until the end of the experiment. At 4 dpi, xenografts were sacrificed and fixed with 4% formaldehyde (Thermo Scientific) and kept in 100% methanol at −20 °C for long-term storage. We designed the test to span 4 days not only due to animal ethics constrains but more importantly to give time to perform immunofluorescence, confocal imaging and analysis in a useful time window for future patient advice.

**Drug administration.** We tested several concentrations of bevacizumab (see Supplementary Fig. 1) and chose 250 µg/mL, ~2× the Cmax (maximum plasma concentration found in patients[43]) as our working concentration. Besides the addition of bevacizumab to the E3 medium at 1 dpi at 250 µg/mL, bevacizumab was added at 100 ng/mL to the cell suspension prior to injection (concentration used in vitro for cell lines[44]). For the intravenous injections, bevacizumab was injected at 100 ng/mL into the pericardia space using fluorescent red dextran to check entry into circulation.

**Bevacizumab-FITC labeling.** Bevacizumab was labeled with N-hydroxysuccinimide-ester fluorescein (Thermo Fisher Scientific) according to manufacturer's instructions.

**Whole-mount immunofluorescence.** Antibodies used: anti-activated Caspase3 (rabbit, Cell Signaling-CST), anti-GFP (mouse, Roche), anti-human HLA-MHC class I subunit (rabbit, AbCAM), and anti-phosphohistone H3 (rabbit, Merck Millipore). Primary antibodies were used at 1:100 and secondary antibodies at 1:400. Nuclei were counterstained with DAPI at 50 µg1mg/mL.

**Imaging and quantification.** All images were obtained in a Zeiss LSM 710 confocal microscope, generally with 5 µm interval. Quantification analysis was performed using ImageJ software. For tumor size, cell counter plugin was used and the number of total DAPI (tumor size) = AVG (3 slices Zfirst, Zmidle, Zlast) × total n° slices/1.5. Mitotic figures and activated Caspase3 were quantified manually along the stack and divided by the tumor size of the respective tumor.

**Metastatic potential quantification.** To distinguish between early and late metastatic events, human TNBC and CRC cell lines were injected into the PVS only (no circ) or into PVS plus circulation (circ). At 1 hpi, xenografts were sorted into these two groups. At 4 dpi, the number of xenografts that had micrometastasis in

the CHT region was determined:

$$\% \text{ Micrometastasis in CHT} = \frac{\#\text{Xenografts at 4 dpi with cell in the CHT}}{\#\text{Total of xenografts analyzed}} \times 100.$$

**Vessel density, vessel infiltration and vessel normalization analysis**. Vessel density was assessed throughout z projections of corresponding images using ImageJ[53] Z Projection tool and the percentage of eGFP fluorescent per tumor was quantified. To analyze vessel infiltration, the superficial slices of the tumor were not considered. For vessel normalization analysis, Hs578T TNBC cells were injected into the PVS of 2dpf *Tg(fli1:eGFP; gata1:DsRed)*. At 4 dpi, the percentage of xenografts presenting erythrocytes inside tumor-related vasculature was determined in two steps: (1) divide the number of xenografts with erythrocytes inside tumor-related vessels by the total number of xenografts analyzed and (2) multiply by 100.

$$\text{Vessel density} = \frac{\text{eGFP area}}{\text{Tumor area}}, \text{ Vessel infiltration} = \frac{\text{eGFP area}}{\text{Core of the tumor}}.$$

**Vessel filament analysis**. Vessels analysis was performed using FIJI/ImageJ. The workflow was automated using a macro which follows a modified version of the workflow delineated by ref. [54].

The analysis is performed on confocal 3D stacks of 1 μm. To minimize the impact of the fish vessels on the analysis, the region of interest was manually selected for each data set. The images were filtered and background subtracted prior to segmentation using Fiji built-in function. To analyze the length and branching properties of the vascular network we performed skeletonization of the image mask using the "Skeletonize 2D/3D" built-in function, while for skeleton analysis used the "Analyze Skeleton" function. Relevant parameters were extracted and statistical analysis was performed.

**Statistics and reproducibility**. Statistical analysis was performed using the GraphPad Prism software version 8. All data were challenged by Shapiro-Wilk and D'Agostino & Pearson normality tests. A Gaussian distribution was only assumed for datasets that pass both normality tests and were analyzed by an unpaired *t*-test with Welch's correction. Datasets without Gaussian distribution were analyzed by unpaired and nonparametric Mann–Whitney test. For metastasis incidence and vessel normalization analysis the Fisher's exact test was used. For vessel infiltration data, the unpaired and nonparametric Kolmogorov–Sminorv test was performed. In addition, for small number of samples, namely the zPDXs analysis, we performed an effect size analysis—Cohen's D with a Hedges' *g* correction (*g*). For all the statistical analysis, *P* value (*P*) is from a two-tailed test with a confidence interval of 95%. Statistical differences were considered significant whenever $P < 0.05$ and statistical output was represented by stars as follows: non-significant (ns) > 0.05, *$P \le 0.05$, **$P \le 0.01$, ***$P \le 0.001$, ****$P \le 0.0001$. All the graphs presented the results as average (AVG) ± standard error of the mean (SEM).

**Reporting summary**. Further information on research design is available in the Nature Research Reporting Summary linked to this article.

## Data availability

Authors can confirm that all relevant data are included in the article and/or its Supplementary information files. Remaining data is available on request from the corresponding author.

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

## Acknowledgements

V. Póvoa for training CRdA on quantification of tumor vessels. We thank the Champalimaud Fish Facility (C. Certal and J. Monteiro) for excellent animal care; we thank the Surgery and Histopathology Units of Champalimaud Clinical Center and Hospital Prof. Doutor Fernando Fonseca. We would like to thank Dr Mireia Castillo-Martin, director of the Champalimaud Foundation Biobank (CFB), and all the CFB team for human specimens procurement. We thank Ana Regalado from Instituto Gulbenkian de Ciência Antibody Facility for the FITC labeling of the bevacizumab antibody. We are grateful to Maria de Sousa for the critical reading of the manuscript. We thank the Champalimaud Foundation, Howard Hughes Medical Institute (HHMI), Congento (LISBOA-01-0145-FEDER-022170, co-financed by FCT/Lisboa2020) and FCT-PTDC/MEC-ONC/31627/2017.

## Author contributions

R.F. and M.G.F. conceptualized the research; R.F. and R.V.M. supervised the research; C.R.A. and R.V.M. performed research; A.P. performed the filament vessel analysis; J.G., C.C., A.A., V.N., M.J.B., M.J.C., J.R. and F.C. provided primary tumor samples; C.C. and J.R. for fruitful discussions; C.R.A. and R.F. wrote the paper. Funding: M.G.F. and R.F.

## Competing interests

The authors declare no competing interests.
