## [Peer Review File · Communications Biology]

Reviewers' comments:

Reviewer #1 (Remarks to the Author):

The manuscript by Almeida et al. describes the application of zebrafish tumor xenografts to analyse the efficacy of Bevacizumab (Avastin, a VEGFA inhibitor) which is used in the clinic to treat cancer patients with highly variable outcomes. By examining tumor growth, cell death, proliferation, metastatic potential and angiogenesis the authors aim to correlate Bevacizumab efficiency in the clinic to the outcomes in zPDX models. The authors suggest that using these zPDX models could be used as a predictive model to determine whether or not Bevacizumab treatment would be efficient for individual patients.

The manuscript examines an important question within the field since it is not well understood why Bevacizumab has such variable outcomes.

While there are some issues that need to be addressed (see below), the study should be well suited for publication with further revision.

1. Since Bevacizumab is administered in the clinic intravenously it would be informative if the authors can investigate the outcomes of an injection of Bevacizumab into the bloodstream of zebrafish embryos with xenografts. Also with regards to the Bevacizumab treatments I would argue that matched data of individual xenografts before and after treatment would be more accurate than comparing two separate groups of xenografts.
2. In line with the above comment I believe that it would be interesting to analyse the functionality of the vasculature that is recruited by the different xenograft models. Are these vessels perfused upon injection of a dye? Are the vessels leaky? What do the cell-cell junctions look like in these vessels? This type of more detailed analysis of the integrity of this recruited vessel network might also explain the metastasis potential of the different xenografted models.
3. The vessels in Hs578T and HT29 xenografts are described to be infiltrating into the xenografts. This however can not be appreciated from the images presented in the main figures as it appears that the vessels could also be covering the outside of the tumors. To strengthen this observation the authors should provide images of single confocal slices in the middle of the tumor mass.
4. Throughout the manuscript it would be helpful if the authors describe how certain experiments were performed. For example, how the xenograft model works (where the cells are injected especially) and how embryos were treated with Bevacizumab. This information is currently detailed in the methods section but since many readers would not be familiar with these type of experiments it would be helpful to guide the reader in this aspect.
5. I believe the claim that using zPDX models to predict clinical outcome for Bevacizumab should be moderated in the text since $n=2$ with a similar outcome is insufficient confirmation of predictability. I understand the availability of PDX material complicates these experiments and I agree that the results are interesting, just not completely evidentiary. Also, in the clinic Bevacizumab would be only a part of the treatment regime that patients receive and therefore outcomes in the clinic are determined by many other factors. Alternatively, the full panel of anti-cancer drugs that a patient is receiving could be tested with or without Bevacizumab to really specify the additive effect of the drug.

Reviewer #2 (Remarks to the Author):

The article by Almeida et al with the title "Zebrafish xenografts as a fast screening platform for bevacizumab impact on tumor growth, angiogenesis and micrometastasis" uses zebrafish embryos as a model organism to analyze the impact of an EMA and FDA approved drug in cancer treatment. The authors use both cancer cell lines, as well as patient derived xenografts for their study. To further strengthen their conclusions, they attempt to correlate available patient data, in which the patients have been treated with bevacizumab, with results achieved in zebrafish, when cells derived from such patients are implanted in this animal model. Although their results seem promising, this part of their research needs to be extended to more patients.

Altogether the manuscript is well written and the research is original and of significance both for the scientific as well as the medical communities.

In general, the authors look at xenografted tumor cells proliferation and apoptosis and draw their main conclusions based on these two parameters.

To study apoptosis the authors use a caspase 3 marker to identify apoptotic cells within their tumor cell population.

To analyze cell proliferation the authors make use of DAPI staining. I have strong concerns about the adequacy of using this method for cell proliferation analysis. Although DAPI staining has been previously reported as a valid method for cell cycle staging (Roukos et al, 2015. Nat. Protocols; 10(2):334-348) the same authors refer to its limitations when being used in thicker structures, as would be the case for zebrafish embryos. Furthermore, studies in which DAPI is used for nuclear content analysis in cancer cells show that there are significant changes in nucleus shape and staining intensity (see Mandelkow et al, 2017. Anticancer research; 37(5):2239-2244). As such, the use of DAPI to analyze cell proliferation in the present study may lead to inaccurate results. More appropriate cell cycle markers (such as PCNA, phospho-Histone 3 or Ki67) should be used, which then can be combined with the information obtained from the DAPI staining. Alternatively, if the authors can demonstrate that that PCNA or pH3 staining correlates well with their DAPI analysis, its use in their current study could be considered valid.

Other minor points that could/should be revisited in the manuscript are the following.

1 - Not all figure panels are mentioned in the text (e.g. Fig 1 a-e). Apart from this, the order of appearance of the figures in the text does not correspond to the order in which they appear in the figure itself.

2 - In figure 1k-o, where the authors attempt to show tumor vascularization, they seem to show a maximum intensity projection. They further corroborate this with some videos of the different stacks. This figure would benefit more from either single z planes at different levels of depth in the tissue (such as surface, and middle) or from a video depicting a 3D projection.

3 - Still on the subject of vascularization, in figure 2n the authors show the changes in vessel density before and after bevacizumab treatment. They base these calculations on % total GFP per tumor area. Considering they are using confocal images, a more elegant way to show these differences would be to perform a filament analysis (some image analysis software are programmed for such analysis). This would provide information regarding vessels distribution within the tumor region, vessels ramifications and thickness. Such data could be more informative for the vascularization studies.

4 - In page 6, lines 3-4 the authors mention that "the most vascularized tumors were also smaller in size". Do they have the correlation analyses to validate this claim? If so, this should be included in the corresponding figure.

5 - In all the figures the author should mention specifically which fish lines they used for the specific experiments (fli:GFP? Casper?). Simply stating they used "2 dpf zebrafish larvae and Tg(fli1:eGFP) larvae" raises the question of which other fish line they are referring to.

6 - It would be more informative if the authors, in the graphs where they use individual dots for each fish, would mark in different colors which of those dots belong to the same technical replicate, since they sometimes show the combined results from 3, 2 or independent experiments. This information

would show the reader how homogeneous and comparable their replicates are.

7 - When the authors quantify micrometastasis formation they mostly look at those in the CHT. They do so by using images acquired with a stereomicroscope. Confocal images would have been more informative, as 3D resolution can be better achieved. Do the authors not think that it would be more valuable to quantify general micrometastasis formation? They could then group this information by regions in the fish. This data could provide better answers to the following questions: capacity of micrometastasis formation for all the different study groups; preferential micrometastasis formation in specific regions for the different groups. Is this preference affected by bevacizumab treatment? Do certain cancer cells metastasize preferably in certain regions, and do these preferences correlate with what has been observed in other models, including human patients? Have the author checked if these micrometastasis have more/less proliferation and/or apoptosis after bevacizumab treatment?

8 - The figure legend of figure 3 should be corrected. The authors say they injected Hs578T, MDA-MB-468, HCT116, SW620 and HT29 cells into the PVS of zebrafish. This should be corrected to Hs578T, MDA-MB-468, HCT116, SW620 or HT29. As it is written now it seems they injected all these cell lines at once in the fish.

9 - Please revise gene nomenclature for fish lines it should be *Tg(fli1:eGFP)* in italic

10 - In the material and methods section the author indicate they tested maximum tolerated drug concentration in the fish larvae. Where is this data shown?

11 - In supplementary figure S1 the authors show the vasculature of a non-injected larva. How many fish have they checked? Did they make an analysis of vasculature formation? This is important has they must account for intra-individual variations.

12 - In table S1 it would also be interesting to include, if known, what is the metastatic potential of each of the cell lines they are using, as this could also influence data interpretation after drug treatment.

Reviewer #3 (Remarks to the Author):

In this paper, the Authors describe the effect of bevacizumab on the growth, vascularization and metastatic potential on human tumor cell lines and patient-derived tumor cells grafted in zebrafish embryos.

The attempt is to set up a personalized medicine-based platform for the rapid prediction of the response of cancer patients to bevacizumab.

Major criticisms:

- 1) The "panel" of cancer cell lines used for the first experiments is represented by just 2 TNBC and 3 CRC cell lines. They do not represent a "panel"! Even worse, for each cancer type, one cell line originated from a tumor metastasis and the other(s) from a primary tumor. Given the tumor heterogeneity, this "panel" does not allow drawing any conclusion related to the cancer type under analysis and the potential differences between primary tumors and their metastases.
- 2) This applies also to the zPDX that are characterized by the extreme heterogeneity of the tumors examined.
- 3) No data are shown about the response to bevacizumab of the tumor cells under analysis (the authors only refer to literature data)
- 4) A major critical point affects the whole research plan. To the best of my knowledge, antibodies given to the fish water will never diffuse into the zebrafish embryo. So all the observations just reflect the effect exerted by bevacizumab on tumor cells during their pre-incubation with the antibody before grafting into the embryo! Is there any evidence that bevacizumab is working during the following days?
- 5) 800-1500 tumor cells are injected into the embryo. Tumor size at the end of the experiments

ranges between approx 300 to 2500 cells, depending on tumor type. Thus, some cell types do not grow or even die following their grafting! This is even more evident for zPDXs where tumor size ranges between 200 and 400 cells!

6) Does the number of injected cells (i.e. the tumor size) affect the results in terms of vascularization and response to bevacizumab?

7) The experiment shown in Figure 4 does not appear very logic. Should not the experimental groups be represented by cells injected in PVS versus cells injected in the blood circulation?

8) The 2 "case reports" are not informative and do not allow to help establishing whether the proposed platform is really predictable of the patient response to therapy

With all the limits pointed out above, the paper is just a list of scattered observations on different cancer cells that do not allow drawing any conclusion.

Champalimaud Foundation

Dear Dr. Anam Akhtar

We would like to thank you and reviewers for the critical and careful reading of our manuscript and the opportunity to address all concerns raised, improving our manuscript.

As you would see from the referees' reports, along with other comments, we would specifically like to see **outcome of Bevacizumab injection into bloodstream of zebrafish embryos with xenografts** and a **comparison before and after treatment of same xenograft as requested by reviewer 1 and other cell cycle marker along with DAPI for proliferation studies as mentioned by reviewer 2**. If the revision process takes significantly longer than six months, we will be happy to reconsider your paper at a later date, as long as nothing similar has been accepted for publication at Communications Biology or published elsewhere in the meantime.

Please find these experiments in **Figure R1**, **Figure R2** and **Supplementary Figure 2**. In addition, we included:

- A filament vessel analysis (Fig. 3)
- Tested vessel functionality (Fig. 3)
- New graphs with the distribution of metastasis in the different organs (Supplementary Fig.6)
- Scatter plot graphs of tumor size vs vessel density (Supplementary Fig.5)
- Absorption/diffusion of the bevacizumab antibody (Supplementary Fig.1)
- MOVIES of Hs578T and HT29 and a montage of the serial slices (Supplementary Fig.4)
- Increased HT29 replicates.
- Show in Figure 2 each experiment replicate pairs in different colors (Fig. 2)

Please find also our answers point by point below.

Reviewer #1

We thank Reviewer #1 for carefully reading our manuscript and raising important points, which we will try to address point by point.

The manuscript by Almeida et al. describes the application of zebrafish tumor xenografts to analyse the efficacy of Bevacizumab (Avastin, a VEGFA inhibitor) which is used in the clinic to treat cancer patients with highly variable outcomes. By examining tumor growth, cell death, proliferation, metastatic potential and angiogenesis the authors aim to correlate Bevacizumab efficiency in the clinic to the outcomes in zPDX models. The authors suggest that using these zPDX models could be used as a predictive model to determine whether or not Bevacizumab treatment would be efficient for individual patients. The manuscript examines an important question within the field since it is not well understood why Bevacizumab has such variable outcomes. While there are some issues that need to be addressed (see below), the study should be well suited for publication with further revision.

1. Since Bevacizumab is administered in the clinic intravenously it would be informative if the authors can investigate the outcomes of an injection of Bevacizumab into the bloodstream of zebrafish embryos with xenografts.

We thank Reviewer#1 for the suggestion. We prepared Bevacizumab at 100ng/ml in the same concentration as used to resuspend the cells prior to injection in our original protocol (Yingmiao Liu et al., 2014, Investigational New Drugs). Daily injection into circulation-bloodstream of Bevacizumab at 100ng/ml lead to an increase of apoptosis (**Figure R1 b**, *P=0.0111) and reduction of vessel infiltration (**Figure R1 f**, *P=0.0161). However, we could not detect a decrease in tumor size or vessel density as in our original protocol (**Figure 2 and 3**). These differences might be due to the different

Bevacizumab concentrations used (in the fish water, we applied 250µg/mL). Nevertheless, our results are quite similar between protocols. Therefore, we would like to argue that, although our setup might not be ideal, it is a compromise of feasibility of an already very labor intensive and difficult protocol.

Figure R1. Injection of PBS or Bevac in circulation-bloodstream since 1hpi.

Human cancer cell line Hs578T was injected into the PVS of 2dpf Tg(*fli1:eGFP*) zebrafish larvae. After 1 hour post-injection (1hpi), zebrafish xenografts were randomly distributed and either PBS or Bevacizumab were injected in circulation. The procedure was repeated every 24h for the following three days. At 4dpi, zebrafish xenografts were sacrificed and analysed by confocal microscopy (a, a', d, d'). The percentage of activated caspase3 (b, *P=0.0111), tumor size (c), total vessel density (e) and vessel infiltration (f, *P=0.0161) were analysed and the outcomes are expressed as AVG±SEM. The number of xenografts analyzed are indicated in the representative images and each dot represents one zebrafish xenograft. Results are from one independent experiment. Statistical analysis was performed using an unpaired *t*-test. Statistical results: (ns)>0.05, *≤0.05, **≤0.01, ***≤0.001, ****≤0.0001. Scale bar, 50 µm. All images are anterior to the left, posterior to right, dorsal up and ventral down.

Also with regards to the Bevacizumab treatments I would argue that matched data of individual xenografts before and after treatment would be more accurate than comparing two separate groups of xenografts.

We performed the requested experiments, starting administration of Bevacizumab only when vessels are present (i.e. 3dpi). To this end, Hs578T cancer cells were injected into the PVS of 2dpf Tg(*fli1:eGFP*) zebrafish larvae. At 3dpi, zebrafish xenografts were randomly distributed and either PBS or bevacizumab was injected into circulation. The procedure was repeated every 24h for the following three days and xenografts were imaged daily with a fluorescent stereoscope (Figure R2). At 7dpi, we compared tumor size and vessel density in relation to the first day of treatment (3dpi). We found that bevacizumab induced a small reduction on tumor size (Figure R2e - % of fish that presented a visible tumor volume reduction). However, we were unable to detect differences on tumor vessel density (Figure R2f). Nevertheless, we found striking that tumors were cleared in 30% of fish treated with Bevacizumab (Figure R2g).

Although we agree these experiments mirror an ideal experimental design, they are difficult to implement causing many xenografts to die during procedure, thus making it impossible to obtain solid results for screening bevacizumab efficacy. Therefore, we would argue that our initial protocol is a good compromise of practicality and feasibility.

Figure R2. Injection of PBS or Bevacizumab in circulation after vessel formation since 3dpi
 Human cancer cell line Hs578T was injected into the PVS of 2dpf *Tg(fli1:eGFP)* zebrafish larvae. At 3dpi, zebrafish xenografts were randomly distributed and either PBS or Bevacizumab were injected in circulation. The procedure was repeated every 24h for the following three days and xenografts were imaged daily. At 7dpi, the tumor size and the vessel density of each xenograft was compared in relation to the first day of treatment (3dpi). At 7dpi, zebrafish xenografts were sacrificed and analysed by confocal microscopy (a-d'''). The percentage of xenografts that presented a reduction of tumor volume (e), vessel density (f) and tumor clearance (g). The number of xenografts analyzed are indicated below the charts. All images are anterior to the left, posterior to right, dorsal up and ventral down.

2. In line with the above comment I believe that it would be interesting to analyse the functionality of the vasculature that is recruited by the different xenograft models. Are these vessels perfused upon injection of a dye? Are the vessels leaky?

We thank Reviewer#1 for the suggestion. Indeed, it has been shown that Bevacizumab can normalize previous abnormal vessels (Carmeliet and Jain, 2011, Nature Reviews Drug Discovery). To test this, rather than using a dextran dye (that led to very inconclusive results in our setting), we made use of a transgenic zebrafish line that labels erythrocytes, since a functional blood vessel should be able to carry erythrocytes.

To this end, we injected Hs578T cancer cells (the most angiogenic cell line) into *Tg(fli1:eGFP; gata1:DsRed)*, where erythrocytes are labelled in red and treated with bevacizumab (see new **Figure 3m-n', o**). Hs578T untreated xenografts exhibited well vascularized tumors. However, we could not find any erythrocytes inside these very abnormal and tortuous vessels. In contrast, upon Bevacizumab treatment, we could observe a clear increase in the number of tumors with red blood cells inside the vessels, suggesting that Bevacizumab was able to normalize vessel function.

What do the cell-cell junctions look like in these vessels?

We thank Reviewer#1 for the suggestion. We performed immunofluorescence for ZO.1 and found that this junctional protein is present in both tumor vessels (bevacizumab-treated and untreated xenografts) (**Figure R3**).

Figure R3. Cell junction staining of tumor-associated vessels. Human cancer cell line Hs578T was injected into the PVS of 2dpf *Tg(fli1:eGFP)* zebrafish larvae. Zebrafish xenografts were treated *in vivo* with bevacizumab and compared with untreated controls. At 4dpi, zebrafish xenografts were imaged by confocal microscopy to assess ZO-1 staining, a marker for cell-cell junctions (**a-b'''**).

3. The vessels in Hs578T and HT29 xenografts are described to be infiltrating into the xenografts. This however can not be appreciated from the images presented in the main figures as it appears that the vessels could also be covering the outside of the tumors. To strengthen this observation the authors should provide images of single confocal slices in the middle of the tumor mass.

We thank Reviewer#1 for the suggestion. We provide movies (**MOVIE#1**, **MOVIE#2**, **MOVIE #3** and **MOVIE#4** of the serial stacks) and also a montage of serial confocal images where it is possible to appreciate the infiltration of vessels into Hs578T and HT29 tumors (see new **Supplementary Figure 4**).

4. Throughout the manuscript it would be helpful if the authors describe how certain experiments were performed. For example, how the xenograft model works (where the cells are injected especially) and how embryos were treated with Bevacizumab. This information is currently detailed in the methods section but since many readers would not be familiar with these type of experiments is would be helpful to guide the reader in this aspect.

We have now included more details in the main text lines 122-124 and also in text lines 153-168.

5. I believe the claim that using zPDX models to predict clinical outcome for Bevacizumab should be moderated in the text since n=2 with a similar outcome is insufficient confirmation of predictability. I understand the availability of PDX material complicates these experiments and I agree that the results are interesting, just not completely evidentiary.

We agree with the reviewer and have now toned-down our claims.

Also, in the clinic Bevacizumab would be only a part of the treatment regime that patients receive and therefore outcomes in the clinic are determined by many other factors. Alternatively, the full panel of anti-cancer drugs that a patient is receiving could be tested with or without Bevacizumab to really specify the additive effect of the drug.

We agree with Reviewer#1 and that was our initial goal. However, due to the limited amount of tissue sample, we were only able to do it for bevacizumab.

Reviewer #2

We thank Reviewer #2 for carefully reading our manuscript and raising important points, which we will try to address point by point.

The article by Almeida et al with the title "Zebrafish xenografts as a fast screening platform for bevacizumab impact on tumor growth, angiogenesis and micrometastasis" uses zebrafish embryos as a model organism to analyze the impact of an EMA and FDA approved drug in cancer treatment. The authors use both cancer cell lines, as well as patient derived xenografts for their study. To further strengthen their conclusions, they attempt to correlate available patient data, in which the patients have been treated with bevacizumab, with results achieved in zebrafish, when cells derived from such patients are implanted in this animal model. Although their results seem promising, this part of their research needs to be extended to more patients. Altogether the manuscript is well written and the research is original and of significance both for the scientific as well as the medical communities. In general, the authors look at xenografted tumor cells proliferation and apoptosis and draw their main conclusions based on these two parameters. To study apoptosis the authors use a caspase 3 marker to identify apoptotic cells within their tumor cell population. To analyze cell proliferation the authors make use of DAPI staining. I have strong concerns about the adequacy of using this method for cell proliferation analysis. Although DAPI staining has been previously reported as a valid method for cell cycle staging (Roukos et al, 2015. Nat. Protocols; 10(2):334-348) the same authors refer to its limitations when being used in thicker structures, as would be the case for zebrafish embryos. Furthermore, studies in which DAPI is used for nuclear content analysis in cancer cells show that there are significant changes in nucleus shape and staining intensity (see Mandelkow et al, 2017. Anticancer research; 37(5):2239-2244). As such, the use of DAPI to analyze cell proliferation in the present study may lead to inaccurate results. More appropriate cell cycle markers (such as PCNA, phospho-Histone 3 or Ki67) should be used, which then can be combined with the information obtained from the DAPI staining. **Alternatively, if the authors can demonstrate that that PCNA or pH3 staining correlates well with their DAPI analysis, its use in their current study could be considered valid.**

We thank Reviewer#2 for the suggestion. We have performed pHH3 immunofluorescence in all tumor xenografts and results are very similar to our mitotic figures quantification, being the exception HCT116 xenografts (see new **Supplementary Figure 2**).

In HCT116 xenografts, we could detect an increase of pHH3 staining upon bevacizumab, but not when we quantified the mitotic figures with DAPI staining. pHH3 labels almost all phases of Mitosis from Prophase to Anaphase (not Telophase - so it labels 3 out of 4 phases) (Low et al, 2008 PMID:18723491). We detect an increase in pHH3 and not in mitotic figures and, given that Prophase is very hard to detect with DAPI, we assume that most probably we missed cells in this stage. Since tumor size reduces upon bevacizumab treatment, these results may suggest a possible arrest in Prophase upon bevacizumab without an increase in cell proliferation. In addition, we also detected an overall increased percentage of pHH3 staining in relation to DAPI mitotic figures in SW620 xenografts, but no difference upon bevacizumab treatment. This discrepancy between DAPI mitotic figures and pHH3 may indicate different cell cycle dynamics between cells.

Other minor points that could/should be revisited in the manuscript are the following.

1 - Not all figure panels are mentioned in the text (e.g. Fig 1 a-e). Apart from this, the order of appearance of the figures in the text does not correspond to the order in which they appear in the figure itself.

We apologize and thank Reviewer#2 – we have corrected accordingly.

2 - In figure 1k-o, where the authors attempt to show tumor vascularization, they seem to show a maximum intensity projection. They further corroborate this with some videos of the different stacks. This figure would benefit more from either single z planes at different levels of depth in the tissue (such as surface, and middle) or from a video depicting a 3D projection.

We thank Reviewer#2 for the suggestion. We provide videos (**MOVIE#1, MOVIE#2, MOVIE #3 and MOVIE#4** of the serial stacks) and also a montage of serial confocal images where it is possible to appreciate the infiltration of vessels into Hs578T and HT29 tumors (**Supplementary Figure 3**).

3 - Still on the subject of vascularization, in figure 2n the authors show the changes in vessel density before and after bevacizumab treatment. They base these calculations on % total GFP per tumor area. Considering they are using confocal images, a more elegant way to show these differences would be to perform a filament analysis (some image analysis software are programmed for such analysis). This would provide information regarding vessels distribution within the tumor region, vessels ramifications and thickness. Such data could be more informative for the vascularization studies.

We thank Reviewer#2 for the suggestion and have now performed a filament analysis. ImageJ analysis revealed that upon bevacizumab the number of branching points reduces but overall vessel length remains similar (see new **Figure 3h-j', k, l**).

4 - In page 6, lines 3-4 the authors mention that "the most vascularized tumors were also smaller in size". Do they have the correlation analyses to validate this claim? If so, this should be included in the corresponding figure.

We thank Reviewer#1 for the suggestion and now include a **Supplementary Figure 5** showing scatter plot and linear regression analysis between tumor size and vessel density, where we can observe that smaller tumors tend to have higher vessel density.

5 - In all the figures the author should mention specifically which fish lines they used for the specific experiments (fli:GFP? Casper?). Simply stating they used "2 dpf zebrafish larvae and Tg(fli1:eGFP) larvae" raises the question of which other fish line they are referring to.

We apologize and thank Reviewer#2 – we clarified all lines used, but in general we used *Tg(fli1:eGFP)*.

6 - It would be more informative if the authors, in the graphs where they use individual dots for each fish, would mark in different colors which of those dots belong to the same technical replicate, since they sometimes show the combined results from 3, 2 or independent experiments. This information would show the reader how homogeneous and comparable their replicates are.

We thank Reviewer#2 for the suggestion and now present **Figure 2** as suggested.

7 - When the authors quantify micrometastasis formation they mostly look at those in the CHT. They do so by using images acquired with a stereomicroscope. Confocal images would have been more informative, as 3D resolution can be better achieved. Do the authors not think that it would be more valuable to quantify general micrometastasis formation? They could then group this information by regions in the fish. This data could provide better answers to the following questions: capacity of micrometastasis formation for all the different study groups; preferential micrometastasis formation in specific regions for the different groups. Is this preference affected by bevacizumab treatment? Do certain cancer cells metastasize preferably in certain regions, and do these preferences correlate with what has been observed in other models, including human patients? Have the author checked if these micrometastasis have more/less proliferation and/or apoptosis after bevacizumab treatment?

We thank Reviewer#2 for the suggestion. We double-checked metastasis in the confocal and the most metastatic lines (MDA-468 and HT29) do metastasize to the CHT but also to the gills, brain, and eyes (see new Supplementary **Figure 6**).

8 - The figure legend of figure 3 should be corrected. The authors say they injected Hs578T, MDA-MB-468, HCT116, SW620 and HT29 cells into the PVS of zebrafish. This should be corrected to Hs578T, MDA-MB-468, HCT116, SW620 or HT29. As it is written now it seems they injected all these cell lines at once in the fish.

We thank Reviewer#2 – we correct it accordingly.

9 - Please revise gene nomenclature for fish lines it should be *Tg(fli1:eGFP)* in italic

We thank Reviewer#2 – we correct it accordingly.

10 - In the material and methods section the author indicate they tested maximum tolerated drug concentration in the fish larvae. Where is this data shown?

We thank Reviewer#2 for bringing this to our attention and now included it in Supplementary Fig 1. We also explain in the main text better our methods (lines 153-168):
To test the effects of bevacizumab, we first tested several concentrations of bevacizumab in the E3 medium, based on maximum plasma concentration (C_{max}) found in patients (90-140µg/mL) (European Medicines Agency, Avastin), which did not induce any mortality (Supplementary Fig. 1a). We chose ~2 times the C_{max} (250µg/mL) as our working concentration, since above this concentration would not be feasible due to lack of antibody availability, since we use the leftovers of the clinic. To further test the absorption/diffusion of the bevacizumab antibody, we conjugated the commercial bevacizumab to FITC. Hs578T cell line was used to validate the protocol, since expresses high levels of VEGF-A (**Supplementary Table 1**) and Hs578T tumors were the most vascularized (**Fig.1k**). At 3dpi, when tumors are established and vasculature developed, Hs578T xenografts were incubated with bevacizumab-FITC for 24h in the E3 medium and imaged by confocal microscopy (**Supplementary Fig. 1b-f'**). As depicted in **Supplementary Fig. 1c-f'**, bevacizumab-FITC is detected specifically in the tumor region and not in the tail of the same fish, demonstrating absorption/diffusion and specific binding to the human tumor cells.

11 - In supplementary figure S1 the authors show the vasculature of a non-injected larva. How many fish have they checked? Did they make an analysis of vasculature formation? This is important has they must account for intra-individual variations.

We checked ~10 non-injected fish. However, since our tumors extrude from the PVS, is quite clear what is the tumor induced vasculature.

12 - In table S1 it would also be interesting to include, if known, what is the metastatic potential of each of the cell lines they are using, as this could also influence data interpretation after drug treatment.

It is known the metastatic potential of these cell lines from mouse and *in vitro* data but not how bevacizumab impacts on the metastatic potential of these cells. We included this information in the **Supplementary Table 1**.

Reviewer#3

We thank Reviewer#3 for carefully reading our manuscript and raising important points, which we have addressed point by point.

In this paper, the Authors describe the effect of bevacizumab on the growth, vascularization and metastatic potential on human tumor cell lines and patient-derived tumor cells grafted in zebrafish embryos. The attempt is to set up a personalized medicine-based platform for the rapid prediction of the response of cancer patients to bevacizumab.

Major criticisms:

1) The “panel” of cancer cell lines used for the first experiments is represented by just 2 TNBC and 3 CRC cell lines. They do not represent a “panel”! Even worse, for each cancer type, one cell line originated from a tumor metastasis and the other(s) from a primary tumor. Given the tumor heterogeneity, this “panel” does not allow drawing any conclusion related to the cancer type under analysis and the potential differences between primary tumors and their metastases.

We apologize and have removed the word “panel”.

2) This applies also to the zPDX that are characterized by the extreme heterogeneity of the tumors examined.

We also removed the word “panel” in this context.

3) No data are shown about the response to bevacizumab of the tumor cells under analysis (the authors only refer to literature data)

We did not perform *in vitro* studies and relied on the published literature.

4) A major critical point affects the whole research plan. To the best of my knowledge, antibodies given to the fish water will never diffuse into the zebrafish embryo. So all the observations just reflect the effect exerted by bevacizumab on tumor cells during their pre-incubation with the antibody before grafting into the embryo! Is there any evidence that bevacizumab is working during the following days?

When we designed this assay, we tested adding Bevacizumab just in cells, just in the E3 medium or in both, and only the later showed significant phenotypes in the tumors.

Nevertheless, in order to specifically test for absorption/diffusion of the antibody into larva, we labeled the Avastin-Bevacizumab with FITC. Hs578T cell line was used to validate the protocol, since expresses high levels of VEGF-A and Hs578T tumors were the most vascularized (**Fig.1k**). At 3dpi, when tumors are established and vasculature developed, Hs578T xenografts (not previously treated with bevacizumab) were incubated with bevacizumab-FITC for 24h in the E3 medium and imaged by confocal microscopy (**Supplementary Fig. 1b-f'**). As depicted in **Supplementary Fig. 1c-f'**, bevacizumab-FITC is detected specifically in the tumor region and not in the tail of the same fish, demonstrating absorption and specific binding to the human tumor cells.

5) 800-1500 tumor cells are injected into the embryo. Tumor size at the end of the experiments ranges between approx 300 to 2500 cells, depending on tumor type. Thus, some cell types do not grow or even die following their grafting! This is even more evident for zPDXs where tumor size ranges between 200 and 400 cells!

Yes, Reviewer#3 is correct in this assessment.

6) Does the number of injected cells (i.e. the tumor size) affect the results in terms of vascularization and response to bevacizumab?

We thank Reviewer#3 for the suggestion and now include a **Supplementary Figure 5** showing the inverse correlation between tumor size and vessel density that we found in our xenografts (linear regression).

7) The experiment shown in Figure 4 does not appear very logic. Should not the experimental groups be represented by cells injected in PVS versus cells injected in the blood circulation?

We apologize if we were not clear, but the setup is similar to Reviewer#3's description.

Overall, we inject in 2 different manners: either in the PVS alone without cells in circulation (similar to mouse subcutaneous) or directly into circulation (similar to mouse experimental metastasis). In the first scenario, in order for tumor cells to efficiently establish a micrometastasis at a distant site, cells have to go through all the metastatic steps (from early to late): invade neighboring tissues, intravasate into blood vessels, survive circulation and extravasate to colonize a distant site. In contrast, in the second scenario (i.e the “experimental metastasis”) tumor cells need to survive in circulation, extravasate and colonize (i.e the later steps). Therefore, although in our short assay we cannot evaluate an “evolutionary metastasis”, we are analyzing its potential (i.e. the ability to perform these different stages).

8) The 2 “case reports” are not informative and do not allow to help establishing whether the proposed platform is really predictable of the patient response to therapy

We agree with Reviewer#3. However, we believe our results show an association of resistance to bevacizumab treatment in the Patients with a resistant phenotype in their matching zPDX. Overall, our results with their small numbers and without the right samples (i.e naïve patients that are going to be treated with bevacizumab) only show the feasibility of the zPDX assay.

REVIEWERS' COMMENTS:

Reviewer #1 (Remarks to the Author):

The revised manuscript by de Almeida et al has addressed most of my concerns. The movies that have been added of the z-stacks are very useful to appreciate changes in vessel infiltration. These type of movies would also strengthen changes in vessel network and/or tumor size that are described in other Figures. There is a clear difference between Movie 1 and 2 (Hs578T tumors) whilst in movies 3 and 4 (HT-29 tumors) there is no clear effect of Bevacizumab on the vasculature and on tumor size. The usage of the gata1 transgenic line has convincingly showed in the image presented that the Bevacizumab treatment improved vessel function. The ZO-1 images are taken at a low magnification/resolution, so it is difficult to judge any differences in localization. The only conclusion that can be made from this staining is that ZO-1 is present in both.

I appreciate that the authors have performed extensive experiments to analyse the effect of Bevacizumab upon the xenografts when injected into the bloodstream. There is an obvious effect on the blood vessel network (angiogenesis) which is potentially not surprising. However, there is no effect on tumor size in Figure R1 but there is in R2. Do the authors have any explanations for this that can be included in the discussion:

- dosage?
- treatment duration?
- independent effect of Bevacizumab on tumor cells when pre-incubated?
- the fact that the vessels become functional/ stable?

I believe this data is very important and should be included in the manuscript along with an extensive discussion of the what might underlie the observed differences.

Reviewer #2 (Remarks to the Author):

The article by Almeida et al with the title "Zebrafish xenografts as a fast screening platform for bevacizumab cancer therapy" uses zebrafish embryos as a model organism to evaluate the impact of bevacizumab in the treatment of different sorts of cancers. I find the new title more suitable to the work presented.

In the new version of the manuscripts the authors have revised and adequately answered to many of my initial concerns.

My main concern with the way cell proliferation had been analyzed has now been addressed. The authors have demonstrated that their method to detect cell proliferation with DAPI staining could be corroborated by specific antibody staining to detect cell division, i.e. pHH3.

I have, however a few other questions.

1. In Lines 125-128 "TNBC and CRC cell lines revealed different basal levels of apoptosis, with representative TNBC cells showing higher level of activated Caspase3 than CRC cells (Fig. 1f-j, u). In contrast, quantification of proliferation shows that CRC representative tumors present higher proliferation rates (Fig. 1f'-j', v).": did the authors test the same parameters obtained similar results in vitro? The main reason for this question being that submitting the cells to microinjection can be a rather stressful process for the cells and different cells lines have a different response to the process. Changes in apoptosis could be also an indication that the cells were no longer as "healthy" as during culture.

2. Line 178-179 "These results may point to an earlier onset of Caspase3 induction or induction of alternative pathways of cell death." Although, generally I agree with the statement done by the authors I think such statement would be stronger if the authors could have demonstrated that some of the cells in these tumors died after treatment. Another possible explanation for their observations

could also be that the cells died earlier after injection and were phagocytosed by the fish's immune cells. A way to overcome this alternate explanation would have been to monitor the injected embryos through time.

3. When the authors discuss the possibility that bevacizumab can normalize vessel function do they know if this also occurs in human patients? Could this be one of the reasons why bevacizumab can also potentiate further development of certain types of cancers?

4. In the sections of "Differential response to bevacizumab in zAvatars" the authors make several statements about a tendency to increase or decrease in metastasis, tumour size, apoptosis and micrometastasis formation in the different zAvatars generated. Such statements often lack statistical support. Interpreting P-values above 0.1 as indicating a tendency towards anything can be misleading, even when presenting the graphs. I agree that the data presented here is interesting, but it would require higher n to correctly interpret the achieved results. In my opinion the authors need to be extremely critical of their results and very careful with their statements in this section.

Finally, as a minor point, through the manuscript I have again encountered some mistakes in the numbering of the figures. For example, in supplementary figure 1 the legend for a, seems to correspond to panel b instead. In the text the authors refer to supplementary figure 5b, but in the figure itself there is only one panel. I recommend a careful review of the figure legends and of the figure panels mentioned in the text with regards to the figures.

Overall I find this article very interesting and valuable as it potentiates the use of zebrafish xenografts as a tool for precision medicine. And provided the authors adapt the text accordingly I support the publication of this work in this journal.

Champalimaud Foundation

Dear Reviewers

We would like to thank you for the critical and careful reading of our manuscript and the opportunity to address all concerns raised, improving greatly our manuscript.

Please find our answers point by point below.

Reviewer #1

I appreciate that the authors have performed extensive experiments to analyse the effect of Bevacizumab upon the xenografts when injected into the bloodstream. There is an obvious effect on the blood vessel network (angiogenesis) which is potentially not surprising. However, there is no effect on tumor size in Figure R1 but there is in R2. Do the authors have any explanations for this that can be included in the discussion:

- dosage? Dosage was the same in R1 and R2
- treatment duration? Duration was the same = 4days
- independent effect of Bevacizumab on tumor cells when pre-incubated? In both settings cells were not pre-incubated with Bevacizumab
- the fact that the vessels become functional/ stable? Bevacizumab exerts its tumoricidal activity either if it is added right after injection (tumor-induced vessels are not formed yet) or at 3dpi (vessels are already formed).

The difference between R1 and R2 is the starting timepoint of Bevacizumab injection, R1 at 1hpi with 4days of treatment and R2 at 3dpi after vessels have been formed also for 4days of treatment. In both settings, Bevacizumab is able to induce its anti-tumor effect. In the R1 setting, Bevacizumab induced a significant increase of apoptosis without tumor size reduction. We believe this is different to our original setting (beva in cells and fish media) probably due to the dosage – in the fish media we have a higher dosage, so we believe if we increase the concentration, we would be able to observe a stronger effect. Also, when we inject the drug in circulation it probably does not reach the tumor immediately, since there are no vessels in the place of injection at the time of injection, and so the drug reaches the tumor later in time and only a part of what was added. In our original protocol, cells have Bevacizumab since the beginning and so Bevacizumab can start immediately exerting its activity. Thus, if we incubate one more day or alternatively if we increase the dosage and maintain the timings we believe we would observe the tumor size reduction phenotype.

R2, is a similar setting - we use the same drug dosage and treatment duration but we modify the starting timepoint of drug injection. At the end of the assay we have less xenografts with tumors and the ones that are present are smaller when injected with bevacizumab. In R2 we are finishing the assay much later in development (9dpf) raising the possibility of some interplay with the immune system that may lead to tumor clearance.

I believe this data is very important and should be included in the manuscript along with an extensive discussion of the what might underlie the observed differences.

We have included this data as Supplementary Figure 6 and discussed as above, highlighted in the manuscript in yellow.

Reviewer #2

The article by Almeida et al with the title "Zebrafish xenografts as a fast screening platform for bevacizumab cancer therapy" uses zebrafish embryos as a model organism to evaluate the impact of bevacizumab in the treatment of different sorts of cancers. I find the new title more suitable to the work presented.

In the new version of the manuscripts the authors have revised and adequately answered to many of my initial concerns.

My main concern with the way cell proliferation had been analyzed has now been addressed. The authors have demonstrated that their method to detect cell proliferation with DAPI staining could be corroborated by specific antibody staining to detect cell division, i.e. pHH3. I have, however a few other questions.

1. In Lines 125-128 "TNBC and CRC cell lines revealed different basal levels of apoptosis, with representative TNBC cells showing higher level of activated Caspase3 than CRC cells (Fig. 1f-j, u). In contrast, quantification of proliferation shows that CRC representative tumors present higher proliferation rates (Fig. 1f'-j', v).": did the authors test the same parameters obtained similar results *in vitro*? The main reason for this question being that submitting the cells to microinjection can be a rather stressful process for the cells and different cells lines have a different response to the process. Changes in apoptosis could be also an indication that the cells were no longer as "healthy" as during culture.

In the work presented here, we did not carry out *in vitro* experiments. However, during the cell staining protocol we performed trypan blue assay to count cells and assess cell viability. We have the percentage of cell death just before cell injection. From all the experiments carried out, we can say that the trend observed *in vivo* is maintained *in vitro*: HCT116 are the ones that show less cell death, whereas Hs578T are the ones that present the highest basal rate of apoptosis. Also, the timings of passages (*in vitro* doublings) are in accordance with our *in vivo* data – so we believe that the relative basal cell intrinsic properties are maintained in our short-time assay.

2. Line 178-179 "These results may point to an earlier onset of Caspase3 induction or induction of alternative pathways of cell death." Although, generally I agree with the statement done by the authors I think such statement would be stronger if the authors could have demonstrated that some of the cells in these tumors died after treatment. Another possible explanation for their observations could also be that the cells died earlier after injection and were phagocyted by the fish`s immune cells. A way to overcome this alternate explanation would have been to monitor the injected embryos through time.

We agree with reviewer#2 and ideally, we could fix several fish at different time points to identify the time pinpoint of cell death. And yes, we agree that tumor cells were most probably phagocyted by the fish`s immune cells.

3. When the authors discuss the possibility that bevacizumab can normalize vessel function do they know if this also occurs in human patients? Could this be one of the reasons why bevacizumab can also potentiate further development of certain types of cancers?

From the literature, it is reported that anti-angiogenic therapies can normalize the vasculature, favouring the delivery of chemotherapy or oxygen for radiotherapy (Batchelor TT 2007, DOI:10.1016/j.ccr.2006.11.021; Emblem 2013 doi:10.1038/nm.3289; Goel 2011 doi: 10.1152/physrev.00038.2010 review; Tonaley 2015 doi: 10.1073/pnas; Sikov 2015 DOI: 10.1200/JCO.2014.57.0572). We totally agree that it is indeed a possibility that if vessels normalize, tumor cells can reach more easily vessels and spread over the body. One of the major drawbacks of these therapies is precisely the disease recurrence by the development of metastasis and drug resistance.

4. In the sections of “Differential response to bevacizumab in zAvatars” the authors make several statements about a tendency to increase or decrease in metastasis, tumour size, apoptosis and micrometastasis formation in the different zAvatars generated. Such statements often lack statistical support. Interpreting P-values above 0.1 as indicating a tendency towards anything can be misleading, even when presenting the graphs. I agree that the data presented here is interesting, but it would require higher n to correctly interpret the achieved results. In my opinion the authors need to be extremely critical of their results and very careful with their statements in this section.

We agree, and indeed due to the reduced amount of sample our zPDX numbers were low, reducing the statistical power of analysis. In order to address this, we performed an additional effect size analysis – Cohen’s D with a Hedges’ g correction for low number of samples (xenografts, see Methods). Nonetheless, we are aware that we cannot take major conclusions only from profiles. And we now state exactly this in the main text (highlighted in yellow).

Finally, as a minor point, through the manuscript I have again encountered some mistakes in the numbering of the figures. For example, in supplementary figure 1 the legend for a, seems to correspond to panel b instead. In the text the authors refer to supplementary figure 5b, but in the figure itself there is only one panel. I recommend a careful review of the figure legends and of the figure panels mentioned in the text with regards to the figures.

We thank Reviewer #2 for alerting to this problem – we have corrected it.

Best regards
Rita Fior